# Non-invasive transdermal delivery of biomacromolecules with fluorocarbon-modified chitosan for melanoma immunotherapy and viral vaccines

Wenjun Zhu[1], Ting Wei [ORCID][1,2], Yuchun Xu[1], Qiutong Jin[1,2], Yu Chao[1], Jiaqi Lu[1,2], Jun Xu[1], Jiafei Zhu[1], Xiaoying Yan[1], Muchao Chen[1], Qian Chen [ORCID][1] ✉ & Zhuang Liu [ORCID][1,2] ✉

Transdermal drug delivery has been regarded as an alternative to oral delivery and subcutaneous injection. However, needleless transdermal delivery of biomacromolecules remains a challenge. Herein, a transdermal delivery platform based on biocompatible fluorocarbon modified chitosan (FCS) is developed to achieve highly efficient non-invasive delivery of biomacromolecules including antibodies and antigens. The formed nanocomplexes exhibits effective transdermal penetration ability via both intercellular and transappendageal routes. Non-invasive transdermal delivery of immune checkpoint blockade antibodies induces stronger immune responses for melanoma in female mice and reduces systemic toxicity compared to intravenous injection. Moreover, transdermal delivery of a SARS-CoV-2 vaccine in female mice results in comparable humoral immunity as well as improved cellular immunity and immune memory compared to that achieved with subcutaneous vaccine injection. Additionally, FCS-based protein delivery systems demonstrate transdermal ability for rabbit and porcine skins. Thus, FCS-based transdermal delivery systems may provide a compelling opportunity to overcome the skin barrier for efficient transdermal delivery of bio-therapeutics.

Transdermal administration refers to needleless drug delivery across the skin without physical damages[1–3]. It has been regarded as an attractive alternative to oral delivery or subcutaneous injection of drugs, due to its unique advantages including non-invasiveness, avoidance of the first-pass effect, painless administration, better patient compliance, avoidance of needle phobia and so on[2,4,5]. Although a variety of transdermal enhancers have been proven to be effective in clinics, the delivered payloads are greatly limited to drugs that have molecular masses around a few hundred Daltons and exhibit strong hydrophobicity[6]. Nowadays, it is still difficult to realize efficient transdermal delivery of hydrophilic

biomacromolecules such as peptides, proteins, or nucleic acids[7]. Besides, the delivery of vaccines is currently one of the hottest research areas in both clinical and scientific communities considering the Coronavirus Disease 2019 (COVID-19) epidemic[8]. Compared to conventional subcutaneous injection or intramuscular injection, transdermal delivery of vaccines may be an attractive approach due to its possibility in at-home administration and the existence of abundant immune cells in the skin[9].

To realize transdermal delivery of biomacromolecule drugs, especially proteins, novel chemical enhancers such as membrane

[1]Institute of Functional Nano & Soft Materials (FUNSOM), Collaborative Innovation Center of Suzhou Nano Science and Technology, Jiangsu Key Laboratory for Carbon-based Functional Materials and Devices, Soochow University, Suzhou 215123, China. [2]Suzhou InnoBM Pharmaceutics Co. Ltd., Suzhou, Jiangsu 215213, China. ✉e-mail: chenqian@suda.edu.cn; zliu@suda.edu.cn

penetrating peptides, as well as various physical enhancement devices, including cavitational ultrasound, electroporation, thermal ablation, microdermabrasion, and microneedles, have been developed[4,10–13]. Although such strategies could be used for transdermal delivery of various macromolecules, including therapeutic proteins, they still face several concerns. For example, the membrane penetrating peptides have been reported to enable transdermal delivery of small proteins such as insulin[14], but with unsatisfactory delivery efficiency[15], and remain to be ineffective for proteins with large molecular weights. Meanwhile, physical enhancement devices such as electroporation and sonophoresis not only can hardly be self-operated, but also could lead to skin damage by high energy pulses[16–19]. Microneedles, which refer to patches with many small needles, have been widely applied in transdermal delivery, showing great potential to deliver insulin and influenza vaccines in recent years[20]. However, the manufacturing process and quality control of microneedle patches, especially with biomacromolecular payloads, would be complicated. Additionally, the microneedles could still induce certain skin damage, which might increase the risk of infections. Some non-invasive platforms such as ionic liquids (ILs) and hyaluronic acids (HAs) were also reported recently to open tight junctions in the stratum corneum and promote paracellular transport[21]. However, they still showed less transdermal efficacy. In several previous studies, lipid nanocarriers such as ethosomes were also reported for the transdermal delivery of proteins against skin tumors[22,23]. Nevertheless, it would still be appealing to develop novel enhancers with high safety and efficiency for transdermal delivery of proteins.

Chitosan (CS) is a biodegradable natural cationic polymer with antibiotic activity and mucoadhesive property[24]. Inspired by the efficient transmucosal efficacy of fluorocarbon-modified CS (FCS), as reported in our previous study for intravesical-instillation-based bladder cancer treatment[25], we speculated that FCS might also be employed for transdermal delivery of biomacromolecules (Fig. 1a).

Herein, we discover that FCS could self-assemble with biomacromolecules such as proteins to form nanocomplexes, which could be added into Aquaphor® as an ointment formulation for topical applications with greatly enhanced transdermal penetration ability. We then employ such FCS-based transdermal delivery platforms for tumor immunotherapy and SARS-CoV-2 vaccines. With the help of FCS, non-invasive transdermal delivery of anti-programmed death-ligand 1(aPDL1) antibody could effectively inhibit the growth of local tumors with direct contact with the FCS/aPDL1-containing ointment. While combined with the co-delivery of anti-cytotoxic T-lymphocyte-associated protein 4 (aCTLA4), such FCS/aPDL1/aCLTA4 ointment could induce strong systemic immune responses to suppress both local and abscopal distant tumors. With significantly enhanced therapeutic responses compared to systemic injection of antibodies at the same dose, our FCS-based local delivery of immune checkpoint antibodies to treat melanoma may lead to fewer concerns in systemic side effects considering the relatively low serum concentrations by the topical administration route. Furthermore, in a proof-of-concept experiment, we verify that FCS could form nanocomplexes with the S1 protein of SARS-CoV-2 as the antigen and polyinosinic: polycytidylic acid (PolyIC), a ligand for toll-like receptor (TLR) 3 as the adjuvant. Topical application of FCS/S1/polyIC nanocomplexes could trigger S1-protein-specific immune responses, reaching a level comparable to that achieved by subcutaneous injection of the same nanocomplex. We further preliminarily demonstrate that FCS-based transdermal delivery may be applicable for skins of larger animals such as rabbits and pigs. Therefore, FCS developed in this work represents a rather effective carrier for transdermal delivery of biomacromolecules, offering possibilities for a wide range of applications such as localized melanoma immunotherapy and self-administered transdermal vaccines against viruses (e.g., SARS-CoV-2).

## Results

### Preparation of FCS/protein nanocomplexes and ex vivo evaluation of their transdermal abilities

FCS was synthesized following a previous report[25]. Briefly, perfluoroalkyl carboxylic acid (PFCA) was grafted to cationic polysaccharide CS through amide coupling at a fluorocarbon substitution of ~4.9% (Fig. S1). Then, FCS was mixed proteins such as immunoglobulin G (IgG) and ovalbumin (OVA) at different mass ratios for 30 min under mild shaking to form nanocomplexes (Fig. 1a). As shown in Fig. 1b, d, both FCS/IgG and FCS/OVA with mass ratio at 1:1 showed sizes around 200 nm in the transmission electron microscopy (TEM) images, consistent with their hydrodynamic diameters measured by dynamic laser light scattering (DLS) (Fig. 1c, e). The zeta potentials (ZP) of both FCS/IgG and FCS/OVA showed high positive charges, which increased from 6.97 to 30.53 mV and 4.38 to 36.73 mV, respectively, as the increase of FCS contents during the formation of nanocomplexes (Fig. 1e). Dynamic light scatter (DLS) measurement (Fig. S2) showed a single peak at ~200 nm for FCS/IgG nano-complexes, which were much larger than the sizes of free IgG, indicating that the majority of IgG should have been encapsulated by FCS. Then, the circular dichroism (CD) spectra were used to verify the structure of proteins before and after forming nanocomplexes. As shown in Fig. 1f, FCS/IgG showed similar CD spectrum characteristic peaks at around 202, 206, and 216 nm to that of free IgG, indicating that the structure of protein remained nearly unchanged during the formation of such FCS-containing nanocomplexes. The intensity difference on 200 nm might result from the solvent. A similar result also was found in the comparison between FCS/OVA and free OVA in Fig. 1g. Additionally, the antibody affinity of aPDL1 in the formulation of FCS/aPDL1 remained nearly unchanged as measured by the competition-enzyme-linked immunosorbent assay (ELISA) (Fig. 1h), further demonstrating that the formation of nanocomplexes wound not affect the activity of contained proteins.

Then, the transdermal kinetics and related mechanisms were investigated. Firstly, the standard Franz diffusion system was used to measure the transdermal delivery efficiency of FCS-containing nanocomplexes across the mouse skin layer. Briefly, fresh skin tissues were fixed between two glass cells, and then FCS/IgG or FCS/OVA, in which IgG and OVA were labeled with fluorescein (FITC), were added into the donor chamber in phosphate-buffered saline (PBS) solutions. The transmitted IgG or OVA was measured by collecting liquid samples in the receptor chamber at different time points to measure the FITC fluorescence (Fig. 1i). All skin resistances were measured pre and post experiment. Skin was regarded as unbroken by the manufactory when its resistance was three times larger than the solvents. In order to avoid destroying the mouse skins, the fat layer was not removed in the mouse skin model. As shown in Figure S3, all skins showed no obvious change in resistance in 12 h. Skins were washed and pyrolyzed to measure the transdermal retention in the dermis region. Transdermal delivery efficiencies of FCS/protein nanocomplexes with different feeding ratios (m/m) were measured at different time points. As shown in Fig. 1i, k, FCS/IgG with the mass ratio at 1:1 showed the highest penetration ability across the skin with the total amount (the sum of permeation and dermis retention) up to about 36 μg/cm² (six times larger than the free group), which may be attributed to the fact that FCS/IgG prepared at 1:1 showed the smallest sizes compared with those prepared at other mass ratios. It could be clearly observed that the zeta potentials of FCS/IgG nanocomplexes increased along with increasing the ratio of FCS, which may be beneficial for skin penetration. However, a further increase of FCS might lead to the aggregation of nanocomplexes. On the other hand, when the amount of IgG increased, the zeta potential of FCS/IgG nanocomplex turned out to be less positive with aggregation, which is also unsuitable for skin penetration. Similarly, FCS/OVA with the mass ratio at 1:1 showed the highest penetration ability with a total penetration (the sum of

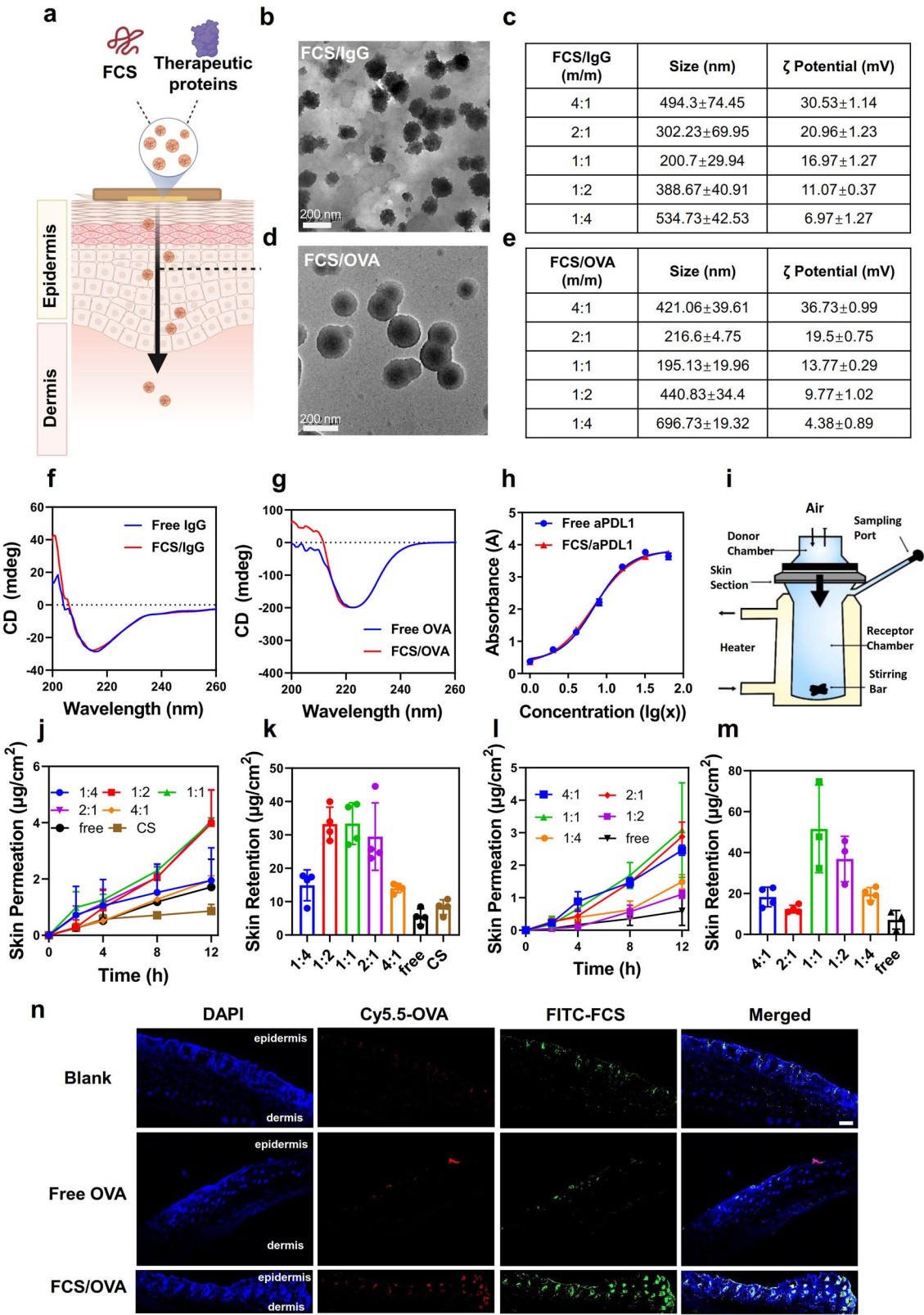

**Fig. 1 | The characterization of FCS-containing nanocomplexes. a** The schematic image of FCS-containing nanocomplexes for transdermal delivery. **b** Representative TEM images of FCS/IgG and **d** FCS/OVA. **c, e** Size distribution and zeta potential of FCS-containing nanocomplexes including **c** FCS/IgG and **e** FCS/OVA ($n = 3$). **f, g** Circular Dichroism (CD) spectra of **f** IgG and **g** OVA pre and post FCS coating. **h** The relative binding affinity of aPDL1 with or without FCS measured by the standard indirect ELISA (iELISA) assay ($n = 2$). **i** Schematic illustration of Franz diffusion cell system used for the skin permeation study. **j** Cumulative permeation and **k** retention of FCS/IgG-FITC and **l, m** FCS/OVA-FITC permeated across the mouse skin after incubation with different FCS-containing formulations over time ($n = 4$ for IgG and $n = 3$ for OVA). Total dosage: 200 μg/cm². **n** Representative confocal images of mice skin treated with FITC-FCS/OVA-Cy5.5 for 12 h ($n = 3$). Scale bar: 200 μm. All illustrations were created with BioRender.com. Data are presented as mean ± standard deviation. Source data are provided as a Source Data file.

permeation and dermis retention) up to about 55 µg/cm² (11 times larger than the free group, Fig. 1l, m). The difference in the penetration behaviors of the two FCS-based nano-complexes might be due to the different physical and chemical properties of the proteins, such as the molecular weight or isoelectric point. Afterward, the skins in the Franz diffusion system post transdermal delivery were sliced for confocal microscopy with FITC-FCS/OVA-Cy5.5 to further evaluate the penetration of FCS nanocomplexes. As shown in Fig. 1n, the colocalization of FCS/OVA was observed in the dermis region of the skin, demonstrating that FCS/OVA could penetrate into the skin dermis in the nanocomplex forms. In contrast, no obvious skin penetration was observed in the free OVA group. Collectively, FCS-based nanocomplexes could successfully deliver proteins into the dermis region of mouse skins.

## Transdermal mechanism of FCS-containing nanocomplexes

Next, we investigated the underlying transdermal mechanism of such FCS-containing nanocomplexes. Firstly, the cytotoxicity of FCS nanocomplexes was measured. As shown in Figs. S4 and S5, no obvious cytotoxicity of FCS and CS was observed for HACAT cells. According to previous literature, there are three classical permeation routes for transdermal delivery, including intercellular, transappendageal, and transcellular routes[26]. Firstly, we studied the intercellular route, by which the nanocomplexes could enlarge the space between epidermis cells and thus pass through them[27]. It has been reported that chitosan and its derivatives could cross epithelial cells by intercellular route[28,29]. During the enlargement of intercellular space, we would expect changes in cell resistance, as well as the expression and allocation of related proteins[30,31]. In this case, human skin epidermis cells HACAT were used to form a cell monolayer and the transepithelial electrical resistance (TEER) between the two sides of the monolayer was monitored (Fig. 2a). As shown in Fig. 2b, the cell single layer was formed 6 days later with stable and high TEER. Interestingly, an obvious decrease of TEER was observed with the addition of FCS/IgG on day 11, indicating the destruction of the cell monolayer and opening of intercellular channels after adding FCS-containing nanocomplexes. More interestingly, the re-increase of TEER was observed 4 h later after the removal of FCS/IgG, and returned to its original level in 12 h, demonstrating that FCS/IgG only temporarily opened the intercellular channel. TEM imaging of the skin also revealed the opening of the tight junctions and enlarged intercellular spaces after topical treatment of the skin with FCS/IgG, in comparison with the normal skin (Figs. 2g and S6). To confirm the enlargement of intercellar spaces, the changes of tight junctions (TJs) related proteins such as zonula occludin (ZO)-1 were further evaluated[32]. With the addition of FCS/IgG, while the total expression of ZO-1 remained nearly unchanged, their continuous distribution was distinctly disturbed, indicating the opening of tight junctions along the interface between cells (Fig. 2c, d). Moreover, the phosphorylation of myosin light chain (MLC), an important parameter for cytoskeletal structure[33], was found to be up-regulated in FCS/IgG treated cells, demonstrating that FCS was able to promote the phosphorylation of myosin light chain to induce the contraction of actin and the rearrangement of the cytoskeleton (Fig. 2e, f).

In addition to the enhanced intercellular bypass permeability of FCS-containing nanocomplexes, the transappendageal pathway, which usually plays an important role in the transport of large and water-soluble drugs through the hair follicles, sweat glands, and sebaceous glands, was also investigated in our experiments[15]. With the counter-staining of Keratin (Krt) 14, the hair follicles and sweat glands in the deep dermis region were labeled. As shown in Fig. 2g (white arrows), it was observed that FCS/IgG colocalized with hair follicles and sweat glands, indicating that the transappendageal pathway also played an important role in FCS-based transdermal delivery systems.

We further studied the transcellular route, which signifies the passage of drugs directly across keratinocytes[34]. Different from normal tissue cells, which decompose drugs in lysosomes, polarized cells such as keratinocytes sometimes might expel drugs through exocytosis[27,35]. Apical exocytosis was used to investigate this phenomenon in vitro. Briefly, HACAT cells were incubated with FCS/IgG-FITC for endocytosis. After 12 h incubation, FCS/IgG-FITC was washed away, and the cells were incubated for another 12 h. Then, the fluorescence of FITC in the supernatant was measured to evaluate the apical exocytosis. As shown in Fig. S7, the apical exocytosis rate of FCS/IgG-FITC from HACAT cells was less than 2%, almost the same compared with that treated with free IgG-FITC, indicating negligible transcellular route was involved. Besides, further evaluation was conducted by the Franz diffusion system. As reported, exocytosis was usually mediated by clathrin. Therefore, skins were treated with 0.1 mg/mL chlorpromazine hydrochloride, the inhibitor for clathrin, for 2 h to compare the penetration ability with or without the clathrin inhibitor. As shown in Fig. S8, the skin retention of FCS/IgG-FITC also showed no clear influence. In conclusion, the transdermal delivery of FCS-containing nanocomplexes mainly relied on the intercellular and transappendageal pathways (Fig. 2i).

Conclusively, FCS, as the derivative of chitosan, could also enlarge cellular space by stimulating the phosphorylation of MLC, a phenomenon also observed for unmodified chitosan[28,29]. The phosphorylation of MLC could then lead to the rearrangement of cytoskeletal structure, transforming tight junction-related protein into the cytoplasm. Afterward, the tight junction between epithelial cells would be opened, and the intercellular space would be enlarged to allow the permeation of our nanocomplexes. On the other side, the unique non-hydrophobic non-hydrophilic properties of fluorocarbon chains would make FCS less sticky when penetrating through various biological barriers[36,37]. Therefore, FCS-based nanocomplexes also showed enhanced permeation via hair follicles through the transappendageal pathway.

## Topical application of FCS/antibody nanocomplexes for melanoma treatment

Melanoma, as one of the most common malignant tumors, especially among Caucasians, develops in the melanocytes located in the bottom layer of the skin's epidermis[38]. For melanoma treatment, immunotherapies, especially immune checkpoint blockade using anti-programmed death-1/its ligand (aPD1/aPDL1) antibodies, have been widely used in clinic[39]. Despite the exciting therapeutic result of using aPD1/aPDL1 antibodies for the treatment of melanoma, there are still many limitations, such as the risk of autoimmune diseases after intravenous injection[40]. Considering that FCS could act as the efficient transdermal delivery carrier for proteins, we thus used it as the transdermal delivery platform to deliver aPDL1 antibody for melanoma treatment (Fig. 3a). It was expected that the transmitted aPDL1 could block the PD1/PDL1 pathway to stimulate cytotoxic T cells and lead to remarkable inhibition of tumors.

For in vivo administration of our FCS-based nanocomplexes, blank ointment (Aquaphor®, mainly petrolatum) was mixed with the solution to keep moisture for a long time. The transparent blank ointment would turn into milk-like ointment when mixed with solutions. The milk-like ointment could be absorbed in 12 h after topical administration for full transdermal delivery. Therefore, the administration was applied only for 12 h for in vivo treatment for a standard applied dosage. To examine the topical penetration behavior of antibodies complexed with FCS, tumors from the mice swiped with ointment containing radioisotope ¹²⁵I labeled IgG (¹²⁵I-IgG) or FCS/¹²⁵I-IgG were collected for quantification of transmitted antibodies at different time points (Fig. S9). As shown in Fig. 3b, compared with free ¹²⁵I-IgG in the transdermally applied ointment, FCS/¹²⁵I-IgG showed dramatically higher accumulation in the tumor, while the radioactivities in other organs appeared to be much lower. Meanwhile, as shown in Fig. 3c, we found that the tumor accumulation of FCS/IgG showed the peaked level at over 120% of injected dose per gram tissue (%ID/g) at 12 h post

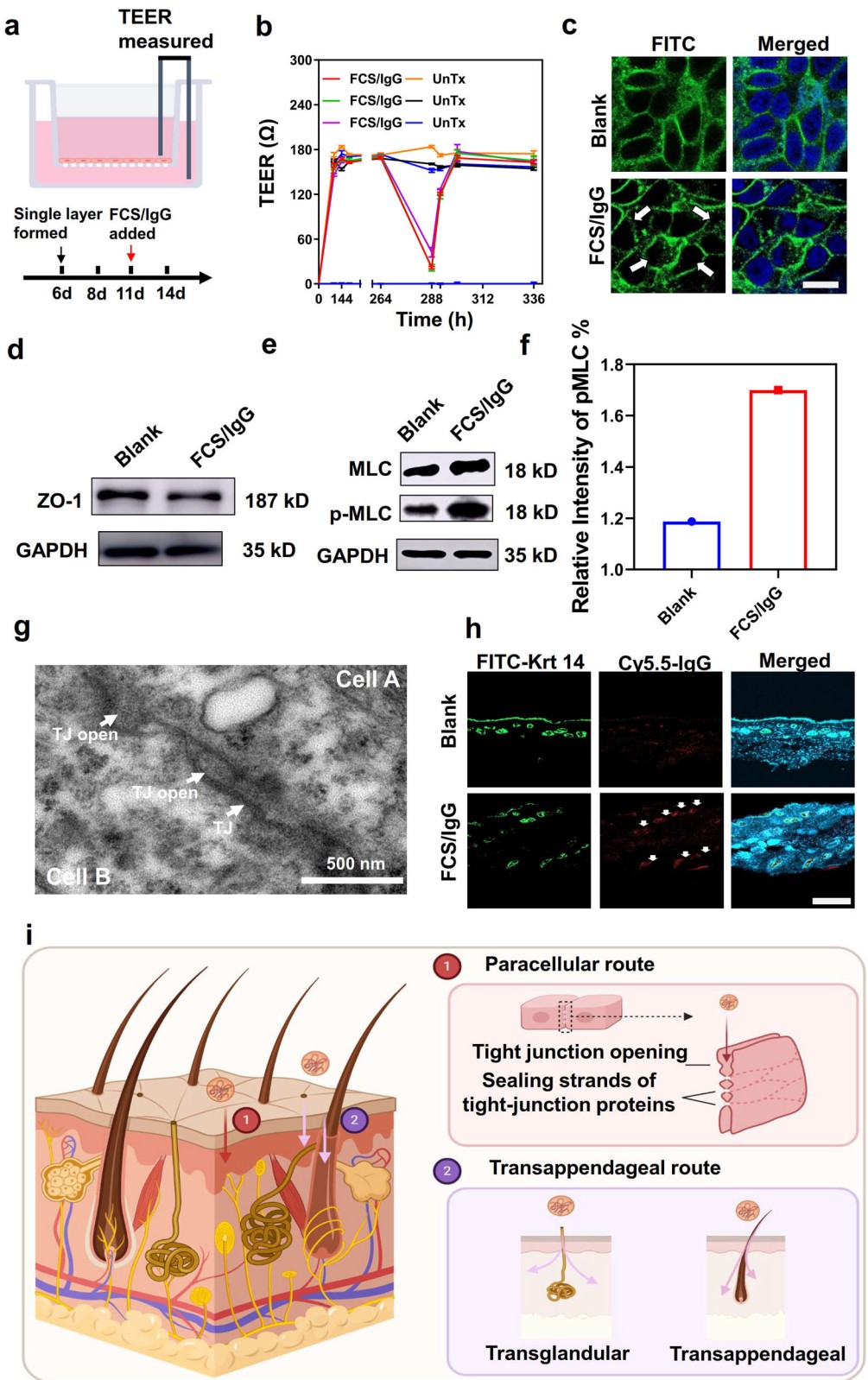

the ointment was applied (about 10% ID in the total tumor in Figure S10). As the ointment was removed at 12 h, the IgG level in the tumor decreased slightly at 24 h. Meanwhile, the ELISA assay was also conducted to detect the tumor accumulation of IgG in the FCS/IgG formulation topically applied to tumors in the ointment. As shown in Fig. S11, the ELISA assay showed similar results with the radiolabeling-based biodistribution data that IgG could be efficiently delivered into

the tumor within 12 h with the aid of transdermal delivery platform using FCS/IgG nanocomplexes. On the other hand, the tumors with FCS/IgG-Cy5.5 were imbedded for tumor slicing. As illustrated in Fig. 3d, the fluorescent signals of IgG-Cy5.5 in the tumor were gradually increased and evenly disbursed inside the whole tumor within 12 h, demonstrating the continuous transmission of antibodies from the FCS/IgG-Cy5.5 ointment into the tumor. Therefore, the accumulation

**Fig. 2 | The transdermal mechanism of FCS-containing nanocomplexes.**
**a** Illustration of the HACAT monolayer cell model. **b** Effects of FCS/IgG on the TEER of the HACAT monolayer cell model ($n = 3$, each TEER was tested 3 times). **c** Immunofluorescence images of the distribution of tight junction-related protein ZO-1 on the HACAT cell membrane after being treated with FCS/IgG ($n = 3$). The white arrows indicated the allocation change of ZO-1. Scale bar: 10 μm. **d, e** Western blotting images showing ZO-1 ($n = 4$) and the phosphorylated level of MLC (p-MLC, $n = 1$) in cells after incubation with FCS/IgG. The raw figures were provided in Figs. S27 and S28. **f** The graphical representations of the relative intensity of MLC/pMLC with the addition of FCS/IgG ($n = 1$). **g** Representative TEM image of skin epithelium after being treated with FCS/IgG. The white arrows indicated the tight junctions (TJs) and the opening of TJs ($n = 3$). **h** Representative immunofluorescence images exhibiting the colocalization of keratin 14 and FCS/IgG-Cy5.5 (white arrows, $n = 3$). Scale bar: 200 μm. **i** The schematic image of the transdermal mechanisms. FCS-containing nanocomplexes could penetrate the skin epidermis through both paracellular and transappendageal routes. By the paracellular route, FCS could stimulate the phosphorylation of MLC and thus open the tight junction between epidermis cells by sealing strands of tight junction proteins. By the transappendageal route, FCS-containing nanocomplexes could cross the epidermis through hair follicles and sweat glands. All illustrations were created with BioRender.com. Data are presented as mean ± standard deviation. Source data are provided as a Source Data file.

of antibody drugs such as aPDL1 in the tumor was evaluated. As shown in Fig. S12, FCS/aPDL1 showed obviously higher tumor accumulation after topical application compared with free and CS/aPDL1. The skin irritation was also evaluated for long-term administration. As shown in Fig. S13, both the film group and the FCS/IgG with film group showed similar skin conditions after topical treatment three times (once every two days), suggesting the safe topical use of FCS.

Inspired by the effective accumulation of IgG in the tumor after transdermal delivery, we then carried out in vivo treatment for melanoma tumors by transdermal delivery of FCS/aPDL1. Mice-bearing melanoma tumors were randomly divided into four groups: (i) Untreated, (ii) Free aPDL1 by intravenous (i.v.) injection, (iii) CS/PDL1 in the ointment by transdermal delivery, and (iv) FCS/aPDL1 in the ointment by transdermal delivery. For transdermal delivery, CS/aPDL1 or FCS/aPDL1 solution was mixed with blank ointment and then applied to the tumor, which was subsequently covered with a 3 M® transparent film. Such treatment was repeated every 2 days three times at the dose of 20 μg aPDL1 per mouse each time. For i.v. injection, 20 μg aPDL1 was administrated into each mouse every 2 days three times. As shown in Fig. 3e, the tumors in the FCS/aPDL1 treated group were successfully inhibited in a short time after the third treatment, while both the CS/aPDL1 group and the free aPDL1 group showed negligible anti-tumor efficacy. During 20 days of observation, the FCS/aPDL1 treated group exhibited the lowest tumor growth rate and the longest survival (Fig. S14).

To understand the immune activation mechanisms of such transdermally delivered immunotherapy, tumors were collected from different groups on day 12 to investigate different types of immune cells, especially T cells, by flow cytometry. As shown in Fig. 3f, for the FCS/aPDL1 treated group, the percentages of both CD4$^+$ and CD8$^+$ T cells showed an obvious increase in the tumor, indicating that aPDL1 was successfully delivered into the tumor to revert the T cell exhaustion. As Granzyme B is important for cell programming death triggered by CD8$^+$ T cells, Ki67 is cell proliferation, and Interferon γ is a cytokine associated with pro-apoptotic and antitumor mechanisms[41], we used these three markers to analyze the activities of CD8+ cells. As can be seen in Fig. 3g–i, all of the three markers in CD8$^+$ T cells was increased in the FCS/aPDL1 group, suggesting the effective infiltration and activation of cytotoxic T lymphocytes (CTLs) in those tumors. For i.v. injection of aPDL1 and transdermal delivery of CS/aPDL1 groups, the tumor infiltrations of both CD4$^+$ and CD8$^+$ T cells, as well as the expressions of granzyme B, Ki67, and IFN-γ, remained nearly unchanged, which might be resulted from the low tumor accumulation of aPDL1 after intravenous injection or transdermal delivery using CS. These results clearly indicated that, compared to i.v. injection of aPDL1, transdermal delivery of aPDL1 using FCS resulted in enhanced antitumor immune responses.

Our platform could also be utilized to co-deliver different therapeutic proteins. In addition to aPDL1, anti-cytotoxic T-lymphocyte-associated protein 4 (aCTLA4) is another important antibody for immune checkpoint blockade to disable regulatory T cells (Tregs) and promote effective T-cell activation[42,43]. Thus, we further investigated the combination therapy using aPDL1 and aCTLA4 co-delivered by the transdermal delivery carrier FCS (Fig. 4a). In this experiment, B16F10 cancer cells were inoculated on the right flank of each mouse as the primary tumor, and a second tumor was inoculated on the opposite side of the same mouse to mimic the cancer metastasis. Three days later, mice bearing two melanoma tumors were randomly divided into five groups: (i) Untreated, (ii) Free aPDL1 and aCTLA4 (i.v.), (iii) FCS/aPDL1, (iv) FCS/aCTLA4, (v) FCS/aPDL1/aCTLA4. All the groups were administrated three times with 20 μg aPDL1 and 20 μg aCTLA4 per mouse each time. It is worth pointing out that after the second i.v. injection of aPDL1 + aCTLA4, half of the mice in this group died, likely due to the horrible side effects (e.g., cytokine storm) triggered by systemic administration of both aPDL1 and aCTLA4. In contrast, the other groups with transdermal delivery of immune checkpoint blockade antibodies showed negligible abnormality. Excitingly, compared to FCS/aPDL1 or FCS/aCTLA4 treated mice, transdermally delivered FCS/aPDL1/aCTLA4 showed further improved therapeutic performance (Fig. 4b, c). More interestingly, the growth of distant tumors in mice with FCS/aPDL1/aCTLA4 treatment also was inhibited (Fig. 4d, e). To understand the abscopal effect induced by FCS/aPDL1/aCTLA4, tumors were collected 12 days after different treatments and analyzed by flow cytometry. As shown in Fig. S19, in the primary tumor, the percentage of CD4+ T cells showed no obvious increase, while that of regulatory T cells was decreased in the primary tumor, which could be the result of the succuss transdermal delivery of aCTLA4. On the other hand, the percentages of both CD8$^+$ T cells and Ki67$^+$CD8$^+$ T cells were increased, illustrating the enhanced tumor infiltration of cytotoxic T lymphocytes (CTLs) (Fig. S20). Consistent with the above results, in the distant tumor, the number of CD8$^+$ T cells, especially Ki67$^+$CD8$^+$ T cells, exhibited an obvious increase in the second tumor (Fig. 4f, i), and the percentages of Tregs were decreased in FCS/aPDL1/aCTLA4 treated group (Fig. 4j).

The excellent anti-tumor efficacy and the increased CTLs in the distant tumor might be attributed to the following mechanisms. Firstly, the activation of CTLs in the local tumor could induce immunogenic death of tumor cells and trigger the chronic exposure of damage-associated molecular patterns (DAMPs) as well as tumor antigens. Antigen-presenting cells (APCs) are then activated and subsequently present tumor antigens to CD8$^+$ T cells, further amplifying systemic antitumor immunity to attack distant tumors. Lastly, as reported, the blockade of PDL1 in the tumor-draining lymph nodes (TDLNs), which may be more efficient to be reached by transdermal delivery, could effectively propel systemic anti-tumor T cell immunity even in distant tumor sites[44,45]. Therefore, our transdermal immune checkpoint antibody delivery could achieve remarkable antitumor efficacies against both local and distant tumors.

## Topical application of FCS/S1/polyIC nanocomplexes for transdermal vaccination

Vaccination is another area of great interest for transdermal delivery[46]. For the transdermal vaccine, in addition to avoiding syringes by health professionals, it could improve immune responses by targeting abundant immune cells beneath the epidermis layer. Since the outbreak of Coronavirus Disease 2019 (COVID-19), various vaccines have

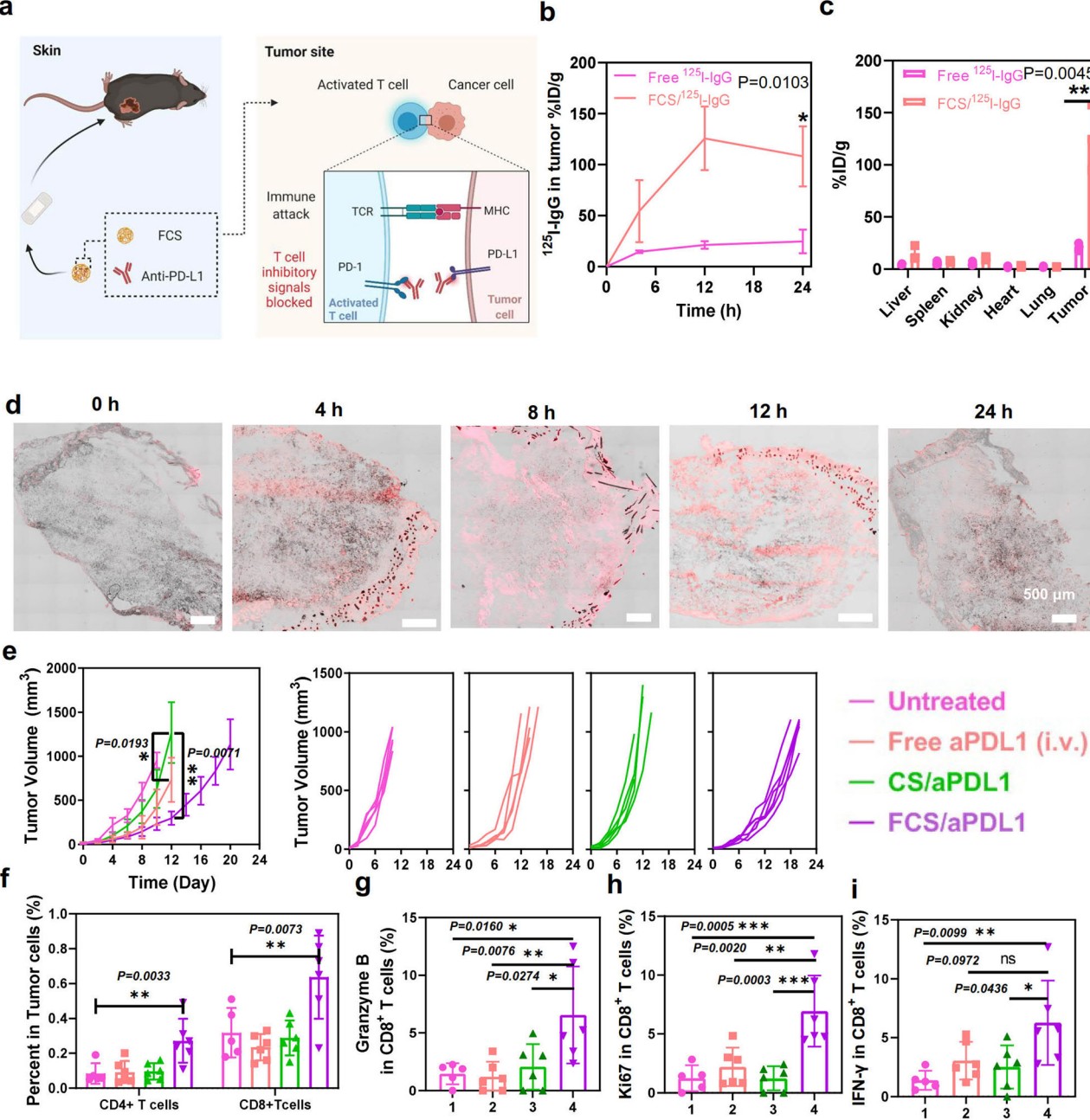

**Fig. 3 | Transdermal delivery of aPDL1 for the treatment of B16F10 melanoma tumors. a** Schematic illusions illustrating the localized transdermal administration of FCS/aPDL1 for the treatment of B16F10 melanoma tumors. **b** The accumulation of FCS/$^{125}$I-IgG in the tumor at different time intervals ($n = 3$). **c** Biodistribution of FCS/$^{125}$I-IgG at 12 h based on radioactivity measurement. The total accumulation and biodistribution analysis was illustrated in Fig. S10 ($n = 3$). **d** Representative confocal images showing the accumulation of FCS/IgG-Cy5.5 in the tumor at different time intervals ($n = 3$). Scale bar: 500 µm. **e** Tumor growth curves of mice in different groups ($n = 5$). Growth curves were stopped when the first mouse in the related group was dead, or its tumor size exceeded 1000 mm³. **f** Quantification of

CD4$^+$ T cells and CD8$^+$ T cells in the tumor after different treatments ($n = 6$). The representative flow cytometric plots were illustrated in Fig. S15. **g–i** Quantification of granzyme B$^+$ (CD3$^+$CD8$^+$Granzyme B$^+$), Ki67$^+$ (CD3$^+$CD8$^+$Ki67$^+$), and IFN-γ$^+$ (CD3$^+$CD8$^+$ IFN-γ$^+$) T cells in the tumor after different treatments ($n = 4$). The representative flow cytometric plots were illustrated in Figs. S16–S18. All illustrations were created with BioRender.com. Data are presented as mean ± standard deviation. Statistical significance was calculated via one-way ANOVA with a Tukey post-hoc test. *$P < 0.05$; **$P < 0.01$; ***$P < 0.001$. Source data are provided as a Source Data file.

been developed to suppress the infection of SARS-CoV-2[47–49]. Up to now, 242 SARS-CoV-2 vaccine candidates are in clinical development, and 11 COVID-19 vaccines have been granted an Emergency Use Listing (EUL) by the WHO[50]. However, most of these vaccines need a subcutaneous or intramuscular injection, which not only requires well-trained medical staff but also faces some additional issues, such as the disposal of a large number of sterile syringes. The transdermal SARS-CoV-2 vaccine, on the other hand, may offer great assistance in

preventing COVID-19 by self-administration, especially in areas with tight medical resources. Considering the rapid global spread of COVID-19 and the efficient transdermal delivery of antibodies using FCS, we further explored whether FCS could be utilized to transdermally deliver the SARS-CoV-2 vaccine in a proof-of-concept study.

To synthesize FCS-based subunit SARS-CoV-2 vaccine, FCS were mixed with the S1 subunit of spike protein of SARS-CoV-2 virus and polyIC, forming the transdermal SARS-CoV-2 vaccine (FCS/S1/polyIC)

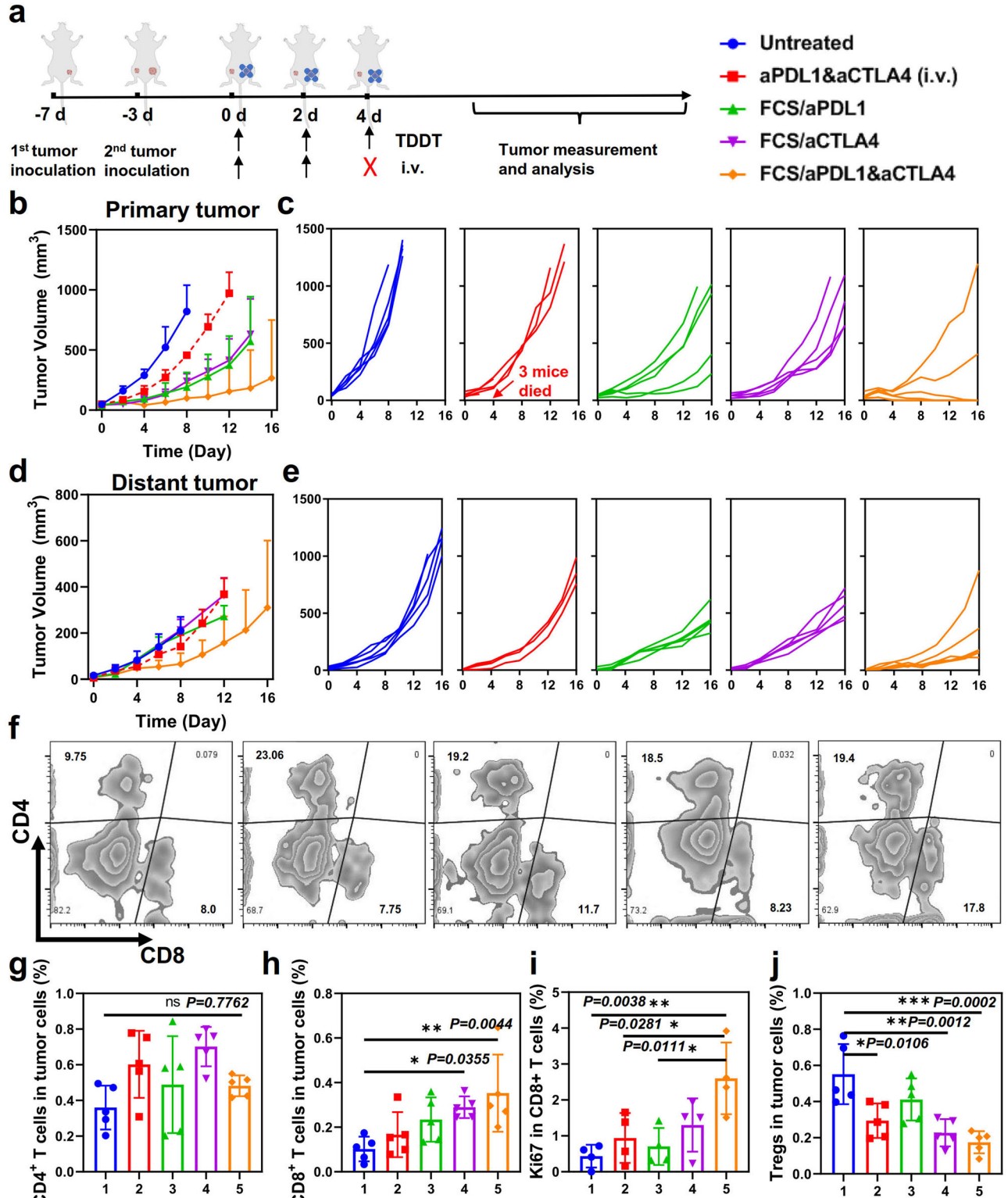

**Fig. 4 | The abscopal effect induced by transdermal delivery of combined immune checkpoint antibodies. a** Schematic illustration of transdermal co-delivery of aPDL1 and aCTLA4 to inhibit the growth of both primary and distant tumors. **b**–**e** Tumor growth curves of primary and distant tumors after different treatments (*n* = 5). Growth curves were stopped when the first mouse in the related group was dead, or the first mouse's tumor size exceeded 1000 mm³. **f**–**h** Representative flow cytometry plots (**f**) and the related quantification of **g** CD4⁺

T cells and **h** CD8⁺ T cells in distant tumors after different treatments (*n* = 5). **i, j** Quantification of Ki67⁺ (CD3⁺CD8⁺Ki67⁺) T cells (**i**) and Tregs (CD3⁺CD4⁺Foxp3⁺) (**j**) in distant tumors after different treatments (*n* = 5). The representative flow cytometric plots were illustrated in Fig. S21. Data are presented as mean ± standard deviation. Statistical significance was calculated via one-way ANOVA with a Tukey post-hoc test. *$P < 0.05$; **$P < 0.01$; ***$P < 0.001$. Source data are provided as a Source Data file.

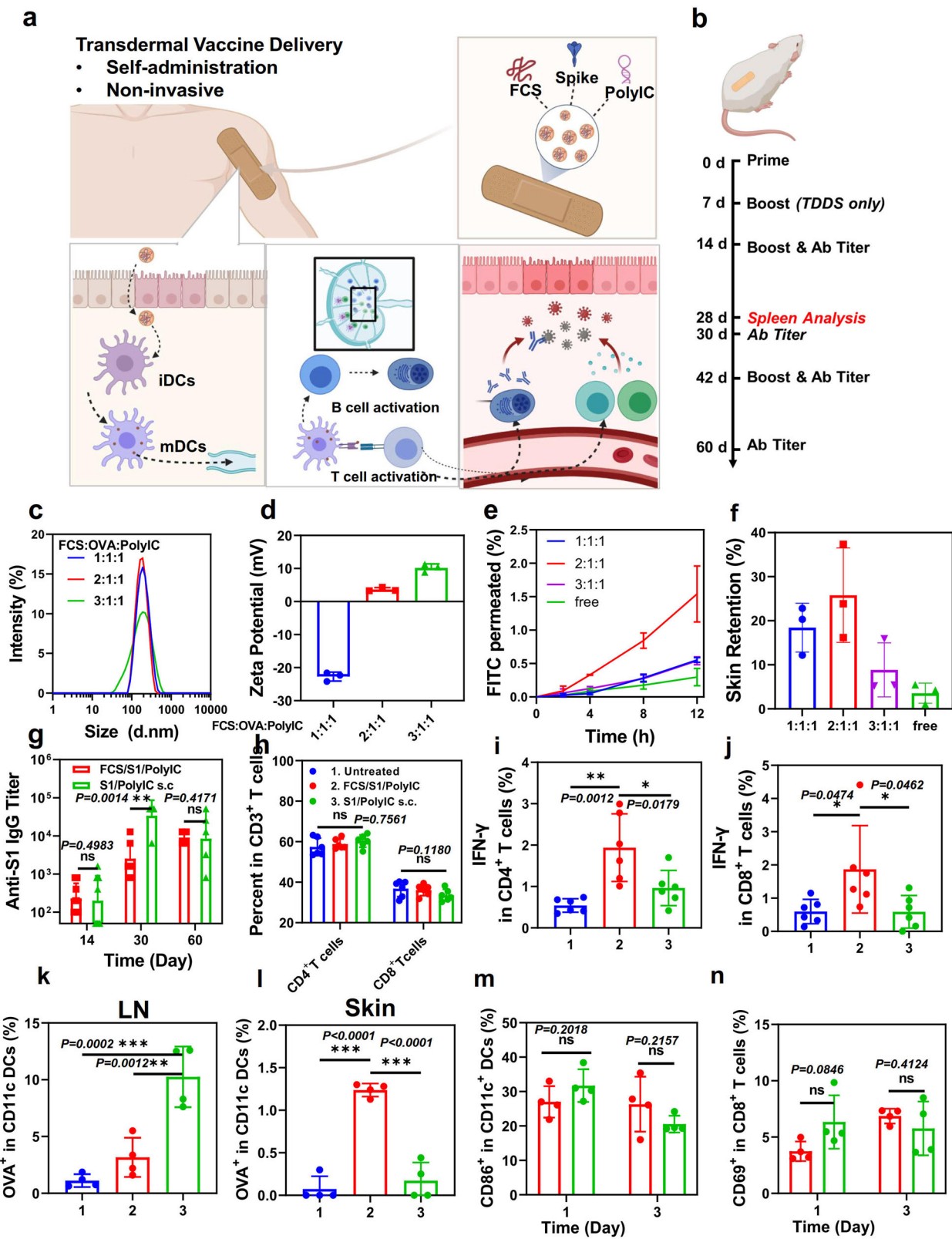

(Fig. 5a). Note that poly IC as a double-strand RNA is a toll-like receptor 3 (TLR 3) agonist that has been commonly used as immune adjuvant[51]. We also optimized the ratio of FCS: antigen: PolyIC in the FCS/S1/ PolyIC vaccine. In this experiment, OVA was used as a modulated antigen for the optimization of the formulation. As shown in Fig. 5c, d, FCS/OVA/polyIC with different mass ratios showed variant sizes at about 200 nm and increased zeta potential with the increase of FCS.

For the skin penetration ability measured by the Franz diffusion system in Fig. 5e, f, FCS/OVA/PolyIC prepared at the mass ratio of 2:1:1 showed the highest skin permeability. This might result from the proper size and zeta potential since nanoparticles prepared at the ratio of 1:1:1 showed dramatically negative charges, while those prepared at the ratio of 3:1:1 showed much larger sizes. Therefore, the formulation with FCS/OVA/PolyIC ratio at 2:1:1 was used for further studies.

**Fig. 5 | Transdermal delivery of SARS-CoV-2 vaccine. a** A schematic illustration for transdermal delivery of the SARS-CoV-2 vaccine and the triggered immune responses. After transdermal delivery, such SARS-CoV-2 nano-vaccines could activate immune cells such as DCs in the dermis, or migrate to the nearby lymph nodes for immune activation. **b** Schematic illustration of the experimental design showing transdermal delivery of SARS-CoV-2 vaccine. **c, d** DLS (**c**) and zeta potential (**d**) of FCS-based transdermal vaccines with different mass ratios from 1:1:1 to 3:1:1 ($n = 3$). **e, f** The skin penetration ability of FCS-based transdermal vaccine with different mass ratios ($n = 3$). Total dosage: 200 μg/cm² (**g**) SARS-CoV-2 specific IgG antibody titers at different time intervals determined by ELISA ($n = 4$). **h** Quantification of CD4+ T cells, CD8+ T cells in the spleen at day 28 ($n = 6$). **i** Quantification of IFN-γ+ secreting CD4+ T cells (CD3+CD4+ IFN-γ+) and **j** CD8+ T cells (CD3+CD8+ IFN-γ+) in the spleen at day 28 ($n = 6$). **k, l** Quantification of OVA-Cy5.5+ (CD45+CD11c+Cy5.5+) in DCs in **k** lymph nodes and **l** skin ($n = 4$). **m, n** Quantification of **m** DC maturation (CD45+CD11c+CD86+) and **n** T cell receptor (TCR) activation (CD45+CD3+CD8+CD69+) in lymph nodes ($n = 4$). All illustrations were created with BioRender.com. The representative flow cytometric plots were illustrated in Figs. S22 & S23. Data are presented as mean ± standard deviation. Statistical significance was calculated via one-way ANOVA with a Tukey post-hoc test. *$P < 0.05$; **$P < 0.01$; ***$P < 0.001$. Source data are provided as a Source Data file.

For the in vivo vaccination experiments, mice were randomly divided into three groups: i. untreated, ii. transdermal delivery of FCS/S1/polyIC, and iii. subcutaneous (s.c.) injection of S1/polyIC. As shown in Fig. 5b, mice in the transdermal delivery group were administrated 3 times in 2 weeks (the doses of S1 and polyIC were both 20 μg per time), while mice in s.c. injection group was injected twice with the S1 protein dose at 20 μg per time and the polyIC dose at 50 μg per time. Interestingly, mice with transdermal delivery of FCS/S1/polyIC showed almost similar antibody titer to that of mice with s.c. injection of S1/polyIC in two weeks, indicating that the transdermal delivery of FCS/S1/polyIC could result in almost the same humoral immunity compared with s.c. administrated vaccines (Fig. 5g). Furthermore, after boosting on day 14, the specific antibody titers in both the transdermal delivery group and s.c. injection group reached up to 10⁴ in 30 days, further indicating the effective activation of humoral immunity by those vaccines. After another boost on day 42, mice were vaccinated by either transdermal delivery of FCS/S1/polyIC or s.c. injection of S1/polyIC showed remained at high levels of anti-S1 antibody titers.

In addition to the activation of humoral immunity by generating specific antibodies against the S1 protein, cell immunity also plays an important role in virus clearance by training cytotoxic T cells to recognize and kill virus-infected host cells[52–54]. Therefore, the levels of cytotoxic T cells in the mouse spleen were evaluated on day 28 after the primeval administration. Although no obvious change of CD4+ and CD8+ T cell infiltration was observed in different groups (Fig. 5h), the secretion of IFN-γ by CD4+ T cells and CD8+ T cells was obviously increased in mice after transdermal delivery of FCS/S1/PolyIC compared to that in mice by s.c. injection of S1/polyIC (Fig. 5i, j), indicating that stronger cytotoxic T-cell responses were induced by transdermal delivery of S1 vaccine. Moreover, the percentage of both effector memory CD4+ and CD8+ T cells (CD3+CD4+CD44+CD62L− and CD3+CD8+CD44+CD62L−) in the spleen of mice treated with FCS/S1/polyIC was also dramatically increased on day 90 with a single boost on day 75 (Fig. S24), while those in mice with subcutaneous administration of S1/PolyIC appeared to be much lower than those in the transdermal group. On day 75 (pre boost) and day 90 (15 days after a single boost), cytokines in the sera of mice after different administrations were measured (Fig. S25). The levels of IL-12p40, an important marker of innate immunity, and IFN-γ, the typical markers of cellular immunity, were all obviously higher in mice treated with FCS/S1/PolyIC, demonstrating that the transdermal delivery of vaccines would trigger long-term adaptive immune memory effect, which might be resulted from the long retention of co-delivered antigen and adjuvant.

In order to investigate the exact permeation ability of vaccines after transdermal administration, we measured the accumulation of antigens in the lymph node 24 h post transdermal delivery or subcutaneous injection of vaccine using Cy5.5-labeled OVA as the modal antigen. As expected, compared to s.c. injection of OVA, mice after transdermal delivery of vaccine after completely swiping the ointment off the skin surface showed lower OVA retention according to the IVIS imaging system (Fig. S26). Then, we further evaluated the uptake of OVA by DCs in lymph nodes and skin (Fig. 5k, l). Interestingly, even though the percentage of OVA+ DCs in the lymph node of mice with s.c. the injection was higher than that of mice with transdermal delivery,

the DCs beneath skins showed higher OVA uptake in mice treated by the ointment containing FCS/OVA/polyIC, revealing that the transdermal delivery of vaccine could directly activate DCs in situ and the activated DCs could migrate to lymph nodes to trigger further immune responses. In contrast, for the s.c. injection group, antigen, and adjuvant would migrate to lymph nodes separately and be more likely to be depleted by non-specific immune cells and activate DCs with low effectiveness[55].

Additionally, we measured the maturation of DC and the activation of CD8+ T cells on one and three days after different treatments. As shown in Fig. 5m, n, the mice with transdermal delivery of vaccines showed almost similar DC maturation and T cell activation to that of mice with s.c. injection, consistent with the observed T cell activation in the spleen. Therefore, although FCS-based transdermal delivery of vaccine showed less absolute antigen penetration compared to that with s.c. injection, it could trigger almost similar humoral immunity and stronger cell immunity. The additional advantages of such FCS-based transdermal vaccines would be the possibility in self-administration and preferable user compliance. However, further studies, such as the SARS-CoV-2 virus challenge, were not conducted due to our current limitations of experimental conditions in handling viruses.

## Penetration ability of FCS-based nanocomplexes on rabbit and porcine models

Next, we further tested the ability of FCS for protein delivery through the skins of larger animals such as rabbits (Fig. 6a). Firstly, the Franz diffusion assay was conducted on rabbit skins in vitro. In this case, the subcutaneous fat layer was removed from rabbit skins for more accurate evaluation. The skin resistances were measured before and after the assay to confirm the skin integrity (Fig. S29). Compared to free IgG, FCS/IgG showed a significant increase in skin permeation for rabbit skins at an amount of about 32 μg/cm² (the sum of permeation and dermis retention, Fig. 6b, c). Vaccination on rabbits was then conducted. Rabbits were randomly divided into three groups: (i) untreated, (ii) topical transdermal application of FCS/OVA/polyIC, and (iii) intramuscular (i.m.) injection of OVA/polyIC. As shown in Fig. 6d, both the transdermal and intramuscular groups were applied with the same dose of OVA and polyIC, following the same administration schedule on day 0, day 14, and day 30. As shown in Fig. 6e, the antibody titer in rabbits of the transdermal vaccination group appeared to be close to that of the i.m. vaccination group, especially at later time points (e.g., days 30 and 60), demonstrating the successful FCS-based transdermal delivery of vaccines in rabbit model for effective vaccination.

Meanwhile, the ex vivo penetration was also conducted on porcine skins. Porcine skins were achieved from the back of little pigs at three months of age and all the subcutaneous fat layer was removed for more accurate evaluation. The skin resistances were measured before and after the assay to confirm the skin integrity (Fig. S30). Similarly, the Franz diffusion assay showed that FCS/IgG also achieved a transdermal ability up to 60 μg/cm² (the sum of proteins penetrated through and retained inside the skin). The larger efficacy might result from the larger hair follicles and sweat glands on porcine. Additionally,

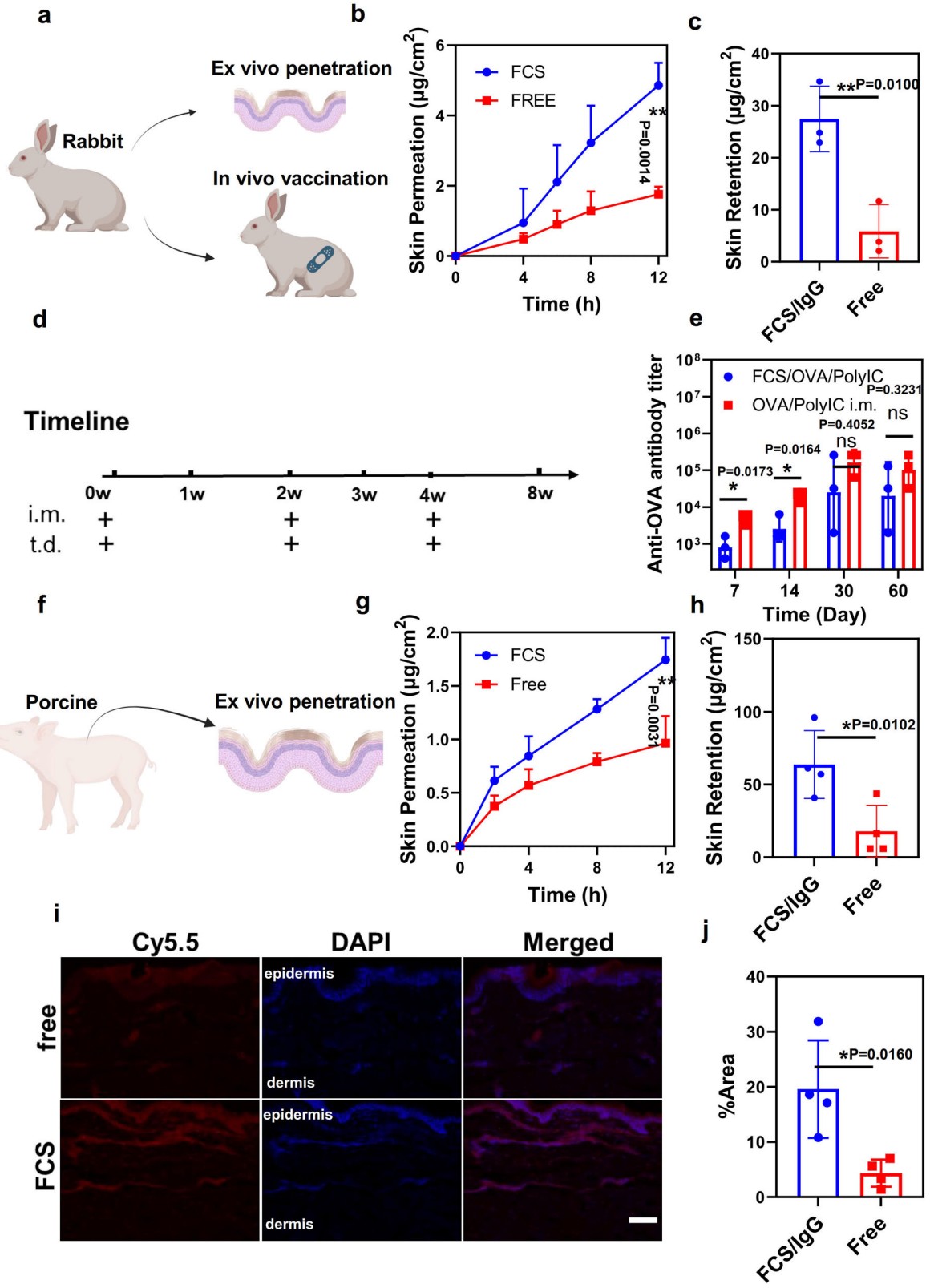

the confocal image of porcine skin slices also showed significant permeation of fluorescently labeled IgG in the dermis region of porcine skins topically applied with FCS/IgG-Cy5.5 (Fig. 6i, j). Collectively, FCS/protein nanocomplexes showed successful transdermal delivery of proteins into the skin dermis and subcutaneous regions for mouse, rabbit, and porcine skins.

## Discussion

In conclusion, we developed a transdermal delivery system based on FCS for efficient local delivery of biomacromolecules, including antibodies, antigens, or nucleic acids (e.g., polyIC), which could be mixed with FCS and added into an ointment for topical applications. For transdermal delivery of antibodies such as aPDL1 and aCTLA4, our

**Fig. 6 | Evaluation of transdermal protein ability on rabbit and porcine models.** **a** Schematic illustration of in vivo vaccination on the rabbit model. **b**, **c** Cumulative percentages of penetration (**b**) and skin retention (**c**) of FCS/IgG-FITC permeated across the rabbit skin over time ($n = 3$). Total dosage: 200 μg/cm². **d** Schematic illustration of the experimental design showing transdermal delivery of OVA vaccination in a rabbit model. **e** OVA-specific IgG antibody titers in rabbit sera at different time intervals determined by ELISA ($n = 3$). **f** Schematic illustration of ex vivo skin penetration on the porcine model. **g**, **h** Cumulative percentages of penetration (**g**) and skin retention (**h**) of FCS/IgG-FITC permeated across the porcine skin over time ($n = 4$). Total dosage: 200 μg/cm². **i** Representative confocal fluorescence images and **j** statistical analysis to show the permeation depth FCS/IgG-Cy5.5 through the porcine skin in 12 h. Free IgG-Cy5.5 was used as a control in those experiments ($n = 4$). Scale bar: 100 μm. All illustrations were created with BioRender.com. Data are presented as mean ± standard deviation. Statistical significance was calculated via one-way ANOVA with a Tukey post-hoc test. *$P < 0.05$; **$P < 0.01$; ***$P < 0.001$. Source data are provided as a Source Data file.

---

FCS-based delivery systems could result in high local antibody accumulation in melanomas and rather strong T cell responses, and thus successfully eliminated primary tumors and inhibited the growth of distant tumors on mice. As reported, the co-delivery of aPDL1 and aCTLA4 by intravenous administration might result in a variety of side effects, including hepatotoxicity, immune-mediated pneumonitis, and even autoimmune endocrinopathies[56]. With significantly enhanced therapeutic responses compared to systemic injection of antibodies at the same dose, our FCS-based local delivery of immune checkpoint antibodies may lead to fewer concerns in systemic side effects considering the relatively low serum concentrations by the topical administration route. On the other hand, transdermal delivery of SARS-CoV-2 vaccines resulted in S1-specific antibody titer similar to that achieved by s.c. injection, and even stronger T-cell responses. Such transdermal delivery of SARS-CoV-2 vaccines might enable fast and vast vaccination even at home once this strategy is proven to be effective in further studies. Such FCS-based transdermal protein delivery is found to be applicable to the skins of larger animals such as rabbits and pigs. Our work thus realized effective noninvasive needleless transdermal delivery of macromolecular therapeutics without the need for any additional physical or chemical stimulations, which has been rather challenging via existing techniques. In addition to delivering immune checkpoint antibodies and vaccines, the macromolecular transdermal delivery carrier developed in this work may also be employed for transdermal delivery of other therapeutic biomacromolecules aiming at diverse medical applications and holds tremendous potential for commercialization.

## Methods

### Ethical statement
All animal studies, including experiments in mice and rabbits, were conducted with the protocols approved by the Soochow University Laboratory Animal Center and the Soochow University Ethics Committee.

### Materials, cell lines, and animal models
Chitosan (DD% ≥95%) was purchased from Aladdin Industrial Co. (Shanghai, China), N-(3-(Dimethylamino)propyl)-N-ethylcarbodiimide hydrochloride crystalline (EDC) and N-hydroxysuccinimide (NHS) were provided by JK Chemical Company (Beijing, China). Phosphate-buffered saline (PBS) was obtained from Beijing Solarbio Science Technology Co., Ltd. SARS-CoV-2 Spike protein was purchased from Sino Biological. Murine melanoma tumor cell line B16F10 cells were cultured under standard conditions recommended by the American Type Culture Collection (ATCC). Female C57BL/6 mice (6-8 weeks) were purchased from Nanjing Pengsheng Biological Technology Co. New Zealand Rabbits (3 kg) were purchased from Suzhou Jinghu Biological Technology Co. Both mice and rabbit skins were collected in the lab. Porcine skins were purchased from Taizhou Taihe Biological Technology Co. All animal studies were conducted with the protocols approved by Soochow University Laboratory Animal Center.

### Synthesis of fluorocarbon-modified chitosan
Perfluoroheptanoic acid (27.5 μmol, 100 mg) was dissolved in dimethyl sulfoxide (DMSO, Sangon Biotech) and then mixed with EDC (78 mg,

1.5 equiv) and NHS (47.43 mg, 1.5 equiv) under stirring in the dark at room temperature for 0.5 h to obtain activated perfluoroheptanoic acid. Then, CS (200 mg) was dissolved in 1% acetic acid solution, mixed dropwise with activated perfluoroheptanoic acid, and stirred in the dark for 12 h to synthesize FCS. The obtained FCS was purified by dialyzing in double-distilled water using a dialysis bag (MWCO 3500 Da, Technologies Co.) for 48 h. Then, the samples were lyophilized, and the degree of substitution of conjugated fluoroalkyl substituents on each CS was characterized by ninhydrin colorimetry following the well-evidenced procedure and analyzed using the microplate reader (Variskan, THERMO). The 19 F NMR (Bruker AV III 600 spectrometers) of FCS (5.0 mg) dissolved in 600 μL of $D_2O$ containing 1.0 mg of $CF_3COOH$ was also applied for the calculation of conjugated fluoroalkyl substituents.

### Preparation of FCS-based nanocomplexes
The synthesized FCS was mixed with IgG in deionized water at different weight ratios (from 1:4 to 4:1) for 0.5 h to prepare FCS/IgG nanocomplexes. FCS/aPDL1, FCS/aCTLA4, FCS/OVA and FCS/S1 were prepared in the same way. The UV−visible absorbance spectra of various complexes were recorded by a PerkinElmer Lambda 750 ultraviolet−visible−NIR spectrophotometer (PerkinElmer, USA). The size, zeta-potential, and dispersibility index of the obtained nanocomplexes were detected by the Zetasizer Nano ZS (Malvern Instrument). Transmission electron microscopy (FEI TECNAI G2) was applied to observe the morphology of representative nanocomplexes.

### Transdermal delivery efficiency
The transdermal delivery ability of FCS was investigated using the Franz diffusion cell system. The Franz diffusion system was purchased from Huanghai Ltd. Shanghai. Fresh mice, rabbits, and porcine back skins were collected and added between the donor and receptor chambers. Samples were collected from the receptor chamber at different time points and the same volume of PBS was added to the receptor chamber to keep the volume unchanged. The final permeation rate was calculated by the formula:

$$\text{Peameation rate}(\%) = (VCn + \sum_{i=1}^{n-1}(ViCi))/A * 100\% \qquad (1)$$

$V$ is the volume of the receptor chamber (8 mL), $Vi$ is the volume of the samples collected at each time point (0.4 mL), $Cn$ is the concentration of FITC in the receptor chamber at each time point, $Ci$ is the concentration of the collected samples at the $n-1$ time point, and A is the feeding concentration of FITC. The feeding concentration of proteins is 1 mg/mL in 400 μL PB solution (pH = 6.5). The permeation skin area is 2 cm².

### Preparation of FCS-based transdermal ointment and in vivo administration
The synthesized FCS/IgG or other FCS/protein nanocomplexes were dropped onto the surface of blank ointment (Aquaphor®, mainly petrolatum) with a mass ratio at 1:1, such as 20 μL FCS/protein with 20 mg ointment. Then, they were gently mixed up with each other. The transparent blank ointment would turn into milk-like ointment, which

was then applied onto the mouse and rabbit skin for transdermal applications.

For mice administration, the mixed milk-like ointment was then swiped on the mouse skin with a round cover of 3.14 cm$^2$ (1 cm in radius) for 12 h. Transparent film (3 M) was covered finally to avoid the influence of mice lick.

For rabbit administration, the mixed milk-like ointment was then swiped on the rabbit skin with a round cover of 50.24 cm$^2$ (4 cm in radius) for 12 h. Transparent film (3 M) was covered finally to avoid the influence of rabbit lick.

### $^{125}$I labeling, stability, and in vivo biodistribution

IgG was labeled with $^{125}$I by the standard chloramine-T oxidation method. Briefly, IgG, Na$^{125}$I, and chloramine-T were mixed at 37 °C for 10 min under slight shaking. The synthesized $^{125}$I-IgG was washed by 10 kDa ultrafiltration three times. To determine radiolabeling stability, $^{125}$I-IgG was mixed with serum for 24 h at 37 °C. Portions of the mixture were sampled at different time intervals and filtered by 10 kDa ultrafiltration. The radioactivities retained on the filters were detected by Wizard2 Gamma Counter (PerkinElmer) to calculate radiolabeling stability.

For in vivo biodistribution of $^{125}$I-IgG assay, B16F10 melanoma tumor-bearing mice were topically administrated with free $^{125}$I-IgG and FCS/$^{125}$I-IgG ointment for different time points. Then, the major organs, including the liver, spleen, kidney, heart, lung, and tumor, were collected and measured by Wizard2 Gamma Counter (PerkinElmer). The cumulation efficacy was calculated by the formulation

$$\%ID/g = R_n/R_0/m_n \times 100\% \qquad (2)$$

Where $R_n$ was the radio intensity of the organ at the exact time point, $m_n$ was the mass of the organ, $R_0$ was the radio intensity of the applied ointment.

### In vivo treatment for melanoma

The B16F10 melanoma tumor model was established by subcutaneous injection of $2 \times 10^6$ B16F10 cells into the right flank of each female C57BL/6 mouse. The mice were randomly divided into five groups when their tumor volumes reached about 50 mm$^3$. Each group was treated with an equal dose of aPDL1 and aCTLA4 (m(ab) = 20 μg each mouse). The tumors of mice were swiped with different FCS ointment for 12 h, and covered by Transparent film (3 M) to avoid the influence of mice lick. The size of the tumor was measured by a caliper every 2 days:

$$\text{Volume} = (\text{length} \times \text{width}^2)/2 \qquad (3)$$

Growth curves were stopped when the first mouse in the related group was dead, or the first mouse's tumor size exceeded 1000 mm$^3$ by ethics from Soochow University Laboratory Animal Center and Soochow University Ethics Committee. In some cases, this limit was exceeded by the last day of measurement, and the mice were immediately euthanized.

### Immunofluorescence staining

For in vitro immunofluorescence staining, HACAT cells were incubated in a 24-well plate to form a single cell layer. Then, FCS/IgG was added and incubated for another 12 h. After removing FCS-IgG, the HACAT cells were fixed by 4% paraformaldehyde, blocked by 5% BSA, and stained with an anti-ZO-1 antibody. One hour later, the cells were stained with a secondary antibody and 4, 6-diamidino-2-phenylindole (DAPI). After mounting, slices were imaged by a confocal fluorescence microscope (Leica SP5). For in vivo immunofluorescence staining, B16F10 tumors and skin were collected from mice at different time points after being swiped with different ointments and embedded in a temperature-responsive OCT gel for immunofluorescence staining with different antibodies, including anti-Keraten 14 and anti-ZO-1. After washing the primary antibodies, the tumor slides were stained with fluorescently labeled secondary antibodies and DAPI. After mounting, the tumor slices were imaged by the confocal fluorescence microscope (Leica SP5).

### Analysis of T cells in different organs

Spleens, tumors, skins, and lymph nodes were collected and homogenized into single-cell suspensions following the standard protocol. Briefly, they were processed through mechanical disruption before digestion for 1 h at 37 °C in an enzymatic solution with RPMI-1640 (10% FBS and 1% PS), 1.5 mg/ml collagenase IV (Sigma), 1.5 mg/ml collagenase I (Sigma), 1.5 mg/ml hyaluronidase (Sigma), and 0.2 mg/ml DNase I (Sigma). The samples were then passed through 200-mesh nylon mesh filters to obtain single-cell suspensions. The obtained single-cell suspensions were incubated with anti-CD16/32 for 30 min at 4 °C. Then, these cells were stained with different antibodies according to the standard protocol.

DCs in lymph nodes and skins were stained with anti-CD11c-FITC (Biolegend, Cat. 117305), anti-CD80-APC (Biolegend, Cat. 104713), and anti-CD86-PE (Biolegend, Cat. 105007).

T cell population in tumors and spleens were stained with anti-CD3-FITC (Biolegend, Cat. 100203), anti-CD4-APC (Biolegend, Cat. 100411), anti-CD45-PerCP (Biolegend, Cat. 103130), and anti-CD8a-PE (Biolegend, Cat. 100707).

To further analyze helper T cells in tumors, the suspension was stained with anti-CD3-FITC (Biolegend, Cat. 100203), anti-CD4-APC (Biolegend, Cat. 100411), anti-Foxp3-PE (Biolegend, Cat. 126403) antibodies with eBioscience™ Foxp3/Transcription Factor Fixation/Permeabilization Concentrate and Diluent according to the manufacturer's protocols. Briefly, after the cells were stained with CD16/CD32 and cell surface markers, they were fixed with the Foxp3 Fixation/Permeabilization working solution for 30 min at room temperature. Then, they were washed with permeabilization buffer at 500$g$ for 5 min and stained with CD16/CD32 in the permeabilization buffer for 30 min at 4 °C. Then, they were stained with anti-Foxp3-PE for another 30 min at room temperature and resuspended in the flow cytometry staining buffer for analysis.

For the analysis of cytotoxic T cell lymphocytes, the suspension was stained with anti-CD3-APC (Biolegend, Cat. 100206), anti-CD8a-PE (Biolegend, Cat. 100707), anti-CD45-PerCP (Biolegend, Cat. 103130) and anti-Ki67-FITC (Biolegend, Cat. 652410) for the level of Ki67 with the same protocol as the Foxp3. The suspension was also stained with anti-CD45-FITC (Biolegend, Cat. 157214), anti-CD8a-PE (Biolegend, Cat. 100707), anti-GranzymeB-PE-Cy7 (Biolegend, Cat. 372214) and anti-CD3-APC (Biolegend, Cat. 100206) for the level of Granzyme B with eBioscience™ Intracellular Fixation & Permeabilization Buffer Set.

For the analysis of IFN-γ + T cells in spleens, the suspension was firstly incubated with a stimulation medium with RPMI-1640 (10% FBS and 1% PS), 50 μM HEPES solution (Sigma), 1 mM Sodium pyruvate solution (Sigma), 50 μM 2-Mercaptoethanol, 10 μg/mL S1 or OVA protein, and 1× eBioscience™ Brefeldin A solution for 6 h and then stained with anti-CD3-FITC (Biolegend, Cat. 100203), anti-CD4-PerCP (Biolegend, Cat. 100432), anti-CD8a-PE (Biolegend, Cat. 100707) and anti- IFN-γ-APC (Biolegend, Cat. 505810) with eBioscience™ Intracellular Fixation & Permeabilization Buffer Set. For the analysis of IFN-γ + T cells in tumors, eBioscience™ Cell Stimulation Cocktail (plus protein transport inhibitors) (500×) was added in the stimulation medium to replace protein and Brefeldin A in the above protocol.

For the analysis of memory T cells, spleens was collected from mice 90 days after the prime-boost and stained with anti-CD3-FITC (eBioscience, Cat. 11-0031), anti-CD8-PerCP-Cy5.5 (eBioscience, Cat. 45-0081), anti-CD44-PE (eBioscience, Cat. 12-0441), and anti-CD62L-

APC (eBioscience, Cat. 17-0621). All these antibodies used in our experiments were diluted by ~200 times.

## S1-specific antibody titer

The S1-specific antibody titer was measured by a standard indirect ELISA (iELISA) method. Briefly, 96-well ELISA plates (Nunc, Thermo) were coated overnight with 1 ng per well of the S1 protein in a coating buffer. The plates were then blocked with 200 µl of ELISA assay buffer (Thermo) and incubated at room temperature for 2 h. Each serum sample was tested in duplicate at a dilution from 1:100 to 1:640,000 twice in ELISA assay buffer (Thermo), in which 100 µl was then added into the wells of each plate for 2 h incubation at room temperature. The horseradish peroxidase (HRP)-conjugated goat anti-mouse IgG (1:10,000, Abcam) was added for 1 h incubation at room temperature. Then, 100 µl TMB solution (Thermo) was added to each well. After 15 min incubation, 100 µl 2 M $H_2SO_4$ solution was added to stop the reaction. The absorbance of each well was read by the 450 nm wavelength (PerkinElmer). The threshold was defied as the double of the average absorbance in the untreated group. The last dilution of a sample that is just larger than the threshold is defined as the antibody titer of this sample.

## Reporting summary

Further information on research design is available in the Nature Portfolio Reporting Summary linked to this article.

## Data availability

The authors declare that all data needed to evaluate the conclusion of this work are within the Article, Supplementary Information, or Source Data file. Source data are provided in this paper.

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

## Acknowledgements

This article was partially supported by the National Research Programs of China (No. 2021YFF0701800, 2020YFA0211100), the National Natural Science Foundation of China (Nos. T2321005, 52032008), the China Postdoctoral Science Foundation funded project (No. 2023M732543), Fundo para o Desenvolvimento das Ciências e da Tecnologia (FDCT 0002/2022/AKP), Emergency Research Project of COVID-19 from Zhejiang University, National Center of Technology Innovation for Biopharmaceuticals, Collaborative Innovation Center of Suzhou Nano Science and Technology, and the 111 Program from the Ministry of Education of China. Z Liu has been supported by the New Cornerstone Science Foundation through the New Cornerstone Investigator Program and the XPLORER PRIZE.

## Author contributions

Wenjun Zhu, Qian Chen, and Zhuang Liu conceived and designed the experiments. Wenjun Zhu and Ting Wei designed and synthesized the materials and nano-complexes. Wenjun Zhu and Qiutong Jin performed the in vitro experiments. Wenjun Zhu, Yuchun Xu and Yu Chao performed the in vivo experiments on melanoma. Wenjun Zhu, Jiaqi Lu, Jun Xu performed the in vivo experiments on vaccines. Wenjun Zhu, Jiafei Zhu and Muchao Chen performed the flow cytometry experiments. Wenjun Zhu, Yuchun Xu and Xiaoying Yan performed all the revised experiments. Wenjun Zhu, Qian Chen and Zhuang Liu analyzed all the data and wrote the paper. All authors discussed the experimental results and edited the paper.

## Competing interests

The authors declare no competing interests.
