## [Peer Review File · Nature Communications]

Non-invasive transdermal delivery of biomacromolecules with
fluorocarbon-modified chitosan for melanoma
immunotherapy and viral vaccinesREVIEWER COMMENTS

Reviewer #1 (Remarks to the Author): with expertise in transdermal drug delivery (also for cancer immunotherapy)

The authors developed a transdermal delivery system based on FCS for efficient local delivery of biomacromolecules. The manuscript is well prepared and impressive. The negative aspects are listed as follows:

(1) Lipid nanocarriers represented by ethosomes have been reported to be effective in transdermal delivery of biological macromolecules and vaccines (e.g. *Journal of Controlled Release*, 2020, 327: 88-99 ; *Acta Biomaterialia*, 2022, 140: 247-260 ;) , which is suggested to be mentioned in the manuscript.

(2) The statement “compared to FCS/aPDL1 or FCS/aCTLA4 treated mice, noteworthy synergistic therapeutic effect was achieved by combination therapy with transdermally delivered FCS/aPDL1/aCTLA4” is not rigorous because the total dosage of the combination group (20 ug + 20 ug) is twice that of the single treatment group. It is insufficient to conclude a synergistic effect unless each single treatment group had a dosage of 40 ug.

(3) The first appearance of MLC should be defined. Western data can only demonstrate changes in MLC expression, not phosphorylation. Therefore, the statement “FCS was able to promote the phosphorylation of myosin light chain...” is not reasonable.

(4) The detection of cytotoxicity of FCS nanocomplexes is not suitable to be included in the section of exploration of transdermal mechanism.

(5) why the penetration of FCS/OVA was much better than that of FCS/IgG? Authors should give some explanation in the text.

(6) On page 4, the in vitro transdermal experiment was not clearly described. In what form is FCS/IgG or FCS/OVA added to the donor chamber? Solution or powder?

(7) Figure 5b should contain an indication of measurement of the antibody titer on the 30th day.

(8) Figure 3 bc lack statistical analysis.

(9) “(j) Cumulative percentage and (k) retention of FCS/IgG-FITC and (l&m) FCS/OVA-FITC permeated across the mouse skin after incubation with different FCS-containing formulations over time” in annotation of Figure 1 is confusing.

(10) On page 3, “Inspired by the outstanding transmucosal efficiency of fluorocarbon modified CS (FCS) as reported in our previous study for intravesical-instillation-based bladder cancer treatment [24],” the citation of this statement is incorrect, should be [25].

(11) Some grammar errors. On page 4, “...FCS was mixed protines...” should be “...mixed with proteins...” ; “...the structure of protein remained nearly unchanged during after formation of such FCS-containing nanocomplexes”, after should be deleted ; “...measured by the competition-enzyme-linked immunosorbent assay (ELISA) (Figure 1h). further demonstrating...” , the full stop should be changed to a comma ;

(12) On page 8, “As another control, aPDL1 was mixed with non-modified chitosan (CS/aPDL1) and added into the cream for topical application”, this sentence is redundant.

Reviewer #2 (Remarks to the Author): with expertise in biomaterials, drug delivery

In the present study, the authors developed a novel method for delivering a promising cancer immunotherapy drug, aPDL1, through the skin to directly treat melanoma tumors. The researchers used a nanocomplex formulation containing aPDL1 and demonstrated that it was effective in reducing tumor growth in mice. The formulation was able to penetrate the skin through paracellular and trans appendageal routes, and the treatment resulted in enhanced antitumor immune responses with no apparent toxicity.

I have reviewed the manuscript and find the research to be valuable and worthy of publication. However, I would like to raise several minor points for your consideration:

1. It is recommended that the authors incorporate the findings and outcomes of the study in the abstract section to provide a concise summary of the research.
2. Could the authors provide information on the influence of the nanoformulation on the enhancement of adaptive immunity? Specifically, have they observed any differences in the expression of CD44/CD62L following the administration of the nanoformulation?
3. The mechanism by which FCS improves transdermal drug delivery has not been mentioned in the manuscript. It would be beneficial to include this information to provide a comprehensive understanding of the underlying mechanisms.
4. Could the authors provide details on the loading capacity of the FCS-based system and the antibody ratio employed in the study? This information would contribute to a better

understanding of the formulation and its effectiveness.

5. Graphical representations of the western blot intensity would greatly enhance the visual presentation of the results. It is recommended that the authors include these graphical representations in the manuscript.

6. Have the authors investigated the toxicity of free chitosan and chitosan-based nanoformulations on normal healthy cells? It would be informative to include any relevant findings or considerations regarding the safety of these formulations.

7. In addition to surface marker analysis, functional assays on memory T cells could provide valuable insights into their effector functions in cancer immunity. These assays can assess the production of cytokines (e.g., interferon-gamma, interleukins) or cytotoxic activity against cancer cells. The authors may consider performing such functional assays to further support their conclusions.

8. Regarding the administration of nanoformulations to mice, it is important to determine the appropriate dosage and establish a standardized dosing regimen. Therefore, I have the following questions:

A. What is the specific dosage of nanoformulations being administered to the mice as part of the experimental procedure?

B. How was the dosing regimen for the nanoformulations standardized to ensure consistency and comparability across treatment groups?

9. It would be valuable to include a discussion on the limitations associated with the use of aPD1/aPDL1 antibodies, particularly the risk of autoimmune diseases after intravenous injection. This information will provide context for the authors' decision to use transdermal delivery rather than intravenous administration. These details can be included in the introduction section.

10. If authors can show the cell cycle analysis in the cells isolated from the tumor, that will ultimately add positives to the article

Additionally, I have a few basic suggestions that will provide additional information to the readers:

11. Could the authors elaborate on how FCS promotes the phosphorylation of myosin light chain, inducing the contraction of actin and rearrangement of the cytoskeleton in HACAT cells?
12. How was the involvement of the transappendageal pathway in FCS-based transdermal delivery determined? It would be beneficial to provide details on the experimental approach or techniques used to assess this pathway.
13. Can the role of the transcellular route in the transdermal delivery of FCS-containing nanocomplexes be further investigated using different experimental approaches or cell models? This could shed more light on the overall transdermal delivery mechanism.
14. Were there any specific physicochemical properties of FCS that contributed to its ability to transport deeply into the hair follicles and permeate through the barriers within the follicles? Providing insights into these properties would enhance our understanding of the nanocomplex behavior.

Thank you for considering these points. I believe that addressing these concerns will enhance the quality and impact of the research article. I look forward to the revised manuscript.

Reviewer #3 (Remarks to the Author): with expertise in transdermal drug delivery (also for cancer immunotherapy)

In this study, fluorocarbon modified chitosan (FCS) for transdermal delivery of immune checkpoint blockade antibodies and SARS-CoV2 antigens was developed, and anti-cancer activity and antibody production were induced by the FCS delivery system. Thus, FCS would be interesting material for transdermal drug delivery. However, important information and data are lacking as follows.

- 1) Reviewer could not image the physicochemical and structural properties of FCS nanoparticles because no chemical structural information about FCS was provided in the manuscript. Therefore, it was difficult to judge whether FCS is a suitable material for transdermal drug delivery.
- 2) Direct evidence of intradermal penetration of FCS nanoparticles should be shown. For example, confocal laser scanning microscopic images of skin section after treatment with FCS nanoparticles including fluorescence dye should be shown. Reviewer could not judge

whether the proteins penetrate into skin as FCS nanoparticles.

3) The phenomenon induced such as tight junction opening by FCS treatment are shown. However, the reason why FCS causes such phenomenon have not been examined.

4) There are two "Figure 5".

5) In later Figure 5 (maybe Figure 6)i, the photographic image should be upside down. In the figure, the stratum corneum (SC) is stained with DAPI, even though the SC has no nuclei. Also, the Cy5.5 and DAPI images of FCS/IgG completely overlap. Therefore, the reviewer could not believe these data.

Reviewer #4 (Remarks to the Author): with expertise in transdermal drug delivery, vaccines, immunology

In this manuscript, the authors present a fluorocarbon-modified chitosan (FCS) that enhances transdermal delivery of therapeutic proteins (antibodies and antigens) and nucleotides (PolyIC) to murine, rabbit, and porcine skin. This study builds on previous work from the group in which they developed the FCS polymer and evaluated its application for transmucosal delivery and treatment of bladder cancer (ref 25). In the current manuscript, the authors nicely demonstrate that this nanocomplex forming material can deliver checkpoint blockade antibodies (anti-PDL1 +/- anti-CTLA4) to treat cutaneous melanomas in mice. They also use the FCS to formulate transdermal vaccines delivering SARS-CoV-2 S1 or OVA antigens plus PolyIC adjuvant to immunize mice and rabbits. This approach and findings, especially with evidence of applicability in multiple animal models for multiple disease indications, would be relevant to a broad audience in the fields of drug delivery and biomaterials. Addressing the following comments will help to improve the manuscript for readers of Nature Communications.

1) Some key information is missing from the methods section. In particular, please add methods regarding the processing (enzymatic dissociation) of skin and tumors for flow cytometric analyses of T cells and DCs. Which Ki67 and IFN γ antibodies, as well as protein transport inhibitors and/or intracellular/intranuclear staining buffers, were used? How were S1-specific antibody titers calculated? The methods only mention measurement of 1:100 dilutions of serum, not titers. How was delivery of ^{125}I -IgG quantified in tumors and

normalized?

2) Clear Granzyme B, Ki67, and IFN γ positive T-cell populations do not appear in the representative flow plots in Figures S12, S13, S14, and S18. How were the positions of these gates determined? Were FMO isotype controls used? Also, what type of ex vivo stimulation, protein transport inhibition, and/or intranuclear or intracellular staining buffers were used? Note that detection of Ki67, a nuclear antigen, requires fix/perm buffers optimized for intranuclear staining (as for FoxP3). Detection of IFN γ typically requires treatment with protein transport inhibitor and antigen-specific (peptide/protein) or non-specific (PMA/Ionomycin) stimulation.

3) In Figure 5i, it would be helpful to include representative fluorescence microscopy images from untreated control skin to confirm that the apparent Cy5.5 signal is not background autofluorescence. Also, while the caption says, "The white line represents the stratum corneum layer (SC)," no such line is apparent in the images.

4) For the Franz diffusion cell experiments used to quantify transdermal delivery in murine, rabbit, and porcine skin (Figures 1j-m, 5b-c, and 5g-h), it would be helpful to include in the figure captions the total dose per area that was applied to the skin, so delivery efficiency can be inferred. From the supplemental methods, it appears that a total dose of 200 $\mu\text{g}/\text{cm}^2$ was applied ($1 \text{ mg}/\text{mL} * 0.4 \text{ mL} / 2 \text{ cm}^2$). Also, since the FCS/protein formulations were applied to skin in PBS solution for the delivery experiments, compared to Aquaphor ointment with occlusive dressing for the in vivo experiments, the authors should briefly discuss how the vehicle and occlusive dressing could influence delivery efficiency and kinetics.

5) In the vaccine experiments, antigen-specificity of the T-cell populations was not evaluated by either peptide-MHC tetramer staining or ex vivo antigen stimulation and intracellular cytokine staining. As such, the reported changes in T-cell populations (IFN γ ⁺ and CD44⁺ CD62L⁻ effector memory T cells) cannot be attributed definitively to antigen-specific T-cell responses. The authors should acknowledge this limitation in the discussion.

6) Page 2, “To realize transdermal delivery of biomacromolecule drugs especially proteins, novel chemical enhancers such as membrane penetrating peptides, as well as various physical enhancement devices including cavitation ultrasound, electroporation, thermal ablation, microdermabrasion and microneedles have been developed 11–16.” – References 11-13 describe transdermal delivery of small molecule drugs, not biomacromolecules. Consider replacing these references with others that focus on transdermal delivery of macromolecules, including: spherical nucleic acids (e.g., Zheng et al. PNAS 2012, DOI: 10.1073/pnas.1118425109) and STAR particles (e.g., Tadros et al. Nat Med 2020, DOI: 10.1038/s41591-020-0787-6).

7) Page 2, “Meanwhile, physical enhancement devices such as electroporation and sonophoresis not only can hardly be self-operated...” – There are multiple recent examples of handheld electroporation devices for cutaneous vaccination, including: Xia et al. PNAS 2021, DOI: 10.1073/pnas.211081711 and Smith et al. Nat Commun 2020, DOI: 10.1038/s41467-020-16505-0.

8) Page 3, “...Aquaphor® as a cream formulation...” – Aquaphor is an ointment, not a cream, which is relevant to the current studies, as ointments and creams, which contain less oil, tend to have different rates of absorption into the skin. References to “cream” here and in the methods should be corrected.

9) Page 4, “As shown in Figure 1f, FCS/IgG showed similar CD spectrum to that of free IgG, indicating that the structure of protein remained nearly unchanged during after formation of such FCS-containing nanocomplexes. Similar result also was found in the comparison between FCS/OVA and free OVA in Figure 1g.” – The CD spectra in Fig 1f and 1g exhibit noticeable differences from ~200-215nm for protein vs protein + FCS formulations. Can the authors comment on the interpretation of these spectra differences?

10) Page 9, “...after the second i.v. injection of aPDL1 + aCTLA4, half of mice in this group died...” – Is this commonly observed in mice for the given dose of aPDL1 and aCTLA4? If so, please provide a reference.

11) Page 11, “Up to now, more than 180 SARS-CoV-2 vaccines are under research

worldwide, and 26 SARS-CoV-2 vaccines are in the stage of clinical trials 48. However, all these vaccines need subcutaneous or intramuscular injection...” – These statistics are outdated (reference 48 is from 2020). There are currently multiple SARS-CoV-2 vaccines in clinical trials which are administered by intranasal route, and a number of microneedle-based COVID-19 vaccines in pre-clinical studies.

12) Pages 11-12, “Interestingly, mice with transdermal delivery of FCS/S1/polyIC showed almost similar antibody titer to that of mice with s.c. injection of S1/polyIC in two weeks, indicating that the transdermal delivery of FCS/S1/PolyIC could result in almost the same B cell activation level compared with s.c. administrated vaccines (Figure 5g).” – Measurement of S1-specific total IgG binding antibody titers alone does not support the claim about “B cell activation levels.” Additional analyses of isotype switching, somatic hypermutation, and affinity maturation, or direct analysis of germinal center B cells would be needed to make this claim.

13) Figures 5g and 6e – Since titers are presented on a log scale, bars and error bars should represent geometric mean and geometric standard deviation, respectively, instead of arithmetic mean and standard deviation or standard error.

14) Statistics are missing in the graphs in Figure 6; however, the corresponding caption has multiple references to statistics and statistical analyses.

15) Survival analysis (Figure S10) should include a statistical analysis (logrank test).

16) In Figure S16, the y-axis for the second row of flow plots should be “FoxP3” instead of “Treg.”

17) Page 4, “...both FCS/IgG and FCS/OVA with mass ratio at 1:1 showed sizes around 100 nm...” – The sizes determined by TEM and DLS appear to be ~ 200 nm.

18) Page 5, “...when the amount of IgG increased, the zeta potential of FCS/IgG nanocomplex turned to be negative with aggregation...” – According to Fig 1c, the lowest

measured zeta potential was still positive (+6.97 mV), so not “negative”, but “less positive.”

19) Page 5, “In this case, human skin dermis cells HACAT were used...” – HaCaT cells are epidermal cells.

20) Figure 1j-m – The figure caption refers to “cumulative percentage”; however, the graphs show “Skin Permeation” and “Retention” in terms of $\mu\text{g}/\text{cm}^2$. Should the caption say “cumulative permeation” instead?

21) Figure 3 caption – “(a) Schematic illustrations illustrating the localized transdermal administration of FCS/aPDL1 & FCS/aCTLA4 for the treatment of B16F10 melanoma tumors.” – FCA/aCTLA4 is not shown in Figure 3a or anywhere in Figure 3.

REVIEWER COMMENTS

Reviewer #1 (Remarks to the Author): with expertise in transdermal drug delivery (also for cancer immunotherapy)

The authors developed a transdermal delivery system based on FCS for efficient local delivery of biomacromolecules. The manuscript is well prepared and impressive. The negative aspects are listed as follows:

1. Lipid nanocarriers represented by ethosomes have been reported to be effective in transdermal delivery of biological macromolecules and vaccines (e.g. Journal of Controlled Release.,2020,327: 88-99; Acta Biomaterialia. 2022,140: 247-260), which is suggested to be mentioned in the manuscript.

Reply: Thank you for the suggestion. The relative discussion has been added in the manuscript as following:

“Some non-invasive platforms such as ionic liquids (ILs) and hyaluronic acids (HAs) were also reported recently to open tight junctions in the stratum corneum and promote paracellular transport.²³ However, they still showed less transdermal efficacy. In several previous studies, lipid nanocarriers such as ethosomes were also reported for the transdermal delivery of proteins against skin tumors.^{24,25} Nevertheless, it would still be appealing to develop novel enhancers with high safety and efficiency for transdermal delivery of proteins.”

2. The statement “compared to FCS/aPDL1 or FCS/aCTLA4 treated mice, noteworthy synergistic therapeutic effect was achieved by combination therapy with transdermally delivered FCS/aPDL1/aCTLA4” is not rigorous because the total dosage of the combination group (20 ug + 20 ug) is twice that of the single treatment group. It is insufficient to conclude a synergistic effect unless each single treatment group had a dosage of 40 ug.

Reply: Thank you for the advice. The word synergistic has been changed in the manuscript as following:

“Excitingly, compared to FCS/aPDL1 or FCS/aCTLA4 treated mice, transdermally delivered FCS/aPDL1/aCTLA4 showed further improved therapeutic performance (Figure 4b & 4c).”

3. The first appearance of MLC should be defined. Western data can only demonstrate changes in MLC expression, not phosphorylation. Therefore, the statement “FCS was able to promote the phosphorylation of myosin light chain...” is not reasonable.

Reply: Thank you for the suggestion. Myosin light chain (MLC) was first defined in the manuscript as following:

“Moreover, the phosphorylation of myosin light chain (MLC), an important parameter for cytoskeletal structure³⁶, was found to be up-regulated in FCS/IgG treated cells, demonstrating that FCS was able to promote the phosphorylation of myosin light chain to induce the contraction of actin and the rearrangement of cytoskeleton (Figure 2e & 2f).”

In Figure 2e, the up-regulation of p-MLC represented the phosphorylation of MLC, the related description has been added in the figure caption as following:

“(d&e) Western blotting images showing ZO-1 and the phosphorylated level of MLC (p-MLC) in cells after incubation with FCS/IgG. The raw figures were provided in Figure S27 & S28.”

4. The detection of cytotoxicity of FCS nanocomplexes is not suitable to be included in the section of exploration of transdermal mechanism.

Reply: We greatly appreciate the reviewer’s suggestion. Hacat cells were used during the study of the mechanism. Therefore, the cytotoxicity study was conducted first to confirm its safety at the cellular level.

5. why the penetration of FCS/OVA was much better than that of FCS/IgG? Authors should give some explanation in the text.

*Reply: Thank you for the advice. The explanation has been added in the text as following:
“The difference in the penetration behaviors of the two FCS-based nano-complexes might be due to the different physical and chemical properties of the proteins such as the molecular weight or isoelectric point.”*

6. On page 4, the in vitro transdermal experiment was not clearly described. In what form is FCS/IgG or FCS/OVA added to the donor chamber? Solution or powder?

*Reply: Thank you for the suggestion. The description of the Franz diffusion system has been added in the manuscript and method part as following:
“Briefly, fresh skin tissues were fixed between two glass cells, and then FCS/IgG or FCS/OVA, in which IgG and OVA were labelled with fluorescein (FITC), were added into the donor chamber in phosphate buffered saline (PBS) solutions.”*

7. Figure 5b should contain an indication of measurement of the antibody titer on the 30th day.

Reply: Thank you for the suggestion. The missing information has been added in Figure 5b.

Figure 5. (b) Schematic illustration of the experimental design showing transdermal delivery of SARS-CoV-2 vaccine.

8. Figure 3 bc lack statistical analysis.

Reply: Thank you for the advice. Statistical analysis has been added in Figure 3 bc.

Figure 3. (b) The accumulation of FCS/ ^{125}I -IgG in the tumor at different time intervals. (c) Biodistribution of FCS/ ^{125}I -IgG at 12 h based on radioactivity measurement. The total accumulation and biodistribution analysis was illustrated in Figure S10.

9. “(j) Cumulative percentage and (k) retention of FCS/IgG-FITC and (l&m) FCS/OVA-FITC permeated across the mouse skin after incubation with different FCS-containing formulations over time” in annotation of Figure 1 is confusing.

Reply: Thank you for the suggestion. The figure caption has been changed as following:

“(j) Cumulative permeation and (k) retention of FCS/IgG-FITC and (l&m) FCS/OVA-FITC permeated across the mouse skin after incubation with different FCS-containing formulations over time.”

10. On page 3, “Inspired by the outstanding transmucosal efficiency of fluorocarbon modified CS (FCS) as reported in our previous study for intravesical-instillation-based bladder cancer treatment [24],” the citation of this statement is incorrect, should be [25].

Reply: Thanks for point it out. The typo error has been corrected.

11. Some grammar errors. On page 4, “...FCS was mixed protines...” should be “...mixed with proteins...”; “...the structure of protein remained nearly unchanged during after formation of such FCS-containing nanocomplexes”, after should be deleted; “...measured by the competition-enzyme-linked immunosorbent assay (ELISA) (Figure 1h). further demonstrating...”, the full stop should be changed to a comma.

Reply: Thank you for the advice. The grammar errors have been corrected.

12. On page 8, “As another control, aPDL1 was mixed with non-modified chitosan (CS/aPDL1) and added into the cream for topical application”, this sentence is redundant.

Reply: Thanks for point it out. The redundant sentence has been deleted.

Reviewer #2 (Remarks to the Author): with expertise in biomaterials, drug delivery

In the present study, the authors developed a novel method for delivering a promising cancer immunotherapy drug, aPDL1, through the skin to directly treat melanoma tumors. The researchers used a nanocomplex formulation containing aPDL1 and demonstrated that it was effective in reducing tumor growth in mice. The formulation was able to penetrate the skin through paracellular and trans appendageal routes, and the treatment resulted in enhanced antitumor immune responses with no apparent toxicity.

I have reviewed the manuscript and find the research to be valuable and worthy of publication. However, I would like to raise several minor points for your consideration:

1. It is recommended that the authors incorporate the findings and outcomes of the study in the abstract section to provide a concise summary of the research.

Reply: Thank you for the suggestion. The finding and outcomes have been added in the abstract as following:

“Thus, FCS-based transdermal delivery systems may provide a compelling opportunity to overcome the skin barrier for efficient transdermal delivery of biomacromolecules including therapeutic antibodies and antigen proteins, widening the range of therapeutics that can be topically administered.”

2. Could the authors provide information on the influence of the nanoformulation on the enhancement of adaptive immunity? Specifically, have they observed any differences in the expression of CD44/CD62L following the administration of the nanoformulation?

Reply: Thanks for point it out. The discussion of the expression of CD44/CD62L has been added in the manuscript as following:

“Moreover, the percentage of both effector memory CD4+ and CD8+ T cells (CD3+CD4+CD44+CD62L- and CD3+CD8+CD44+CD62L-) in the spleen of mice treated with FCS/S1/polyIC was also dramatically increased on day 90 with a single boost on day 75 (Figure S24), while those in mice with subcutaneous administration of S1/PolyIC appeared to be much lower than those in the transdermal group. On day 75 (pre boost) and day 90 (15 days after a single boost), cytokines in sera of mice after different administrations were measured (Figure S25). The levels of IL-12p40, an important marker of innate immunity, and IFN- γ , the typical markers of cellular immunity were all obviously higher in mice treated with FCS/S1/PolyIC, demonstrating that the transdermal delivery of vaccines would trigger long-term adaptive immune memory effect, which might be resulted from the long retention of co-delivered antigen and adjuvant.”

Figure S24. The long-term adaptive immune effect of mice after different vaccination administrations. Splenic lymphocytes were collected 90 days post administrations and stained with FITC-CD3, PE-CD44, APC-CD62L and PERCP-CD4 or PERCP-CD8. The CD44⁺CD62L⁻ cells in both CD4⁺ and CD8⁺ T cells are effector memory CD4⁺ and CD8 T⁺ cells, respectively. Data are presented as mean \pm standard deviation ($n=4$). Statistical significance was calculated via one-way ANOVA with a Tukey post-hoc test. * $P < 0.05$; ** $P < 0.01$; *** $P < 0.001$.

Figure S25. The sera cytokine level of IL-12p40 and IFN- γ (a&b) measured on day 75 (pre boost) and (c&d) day 90. Data are presented as mean \pm standard deviation ($n=3$ or 4). Statistical significance was calculated via one-way ANOVA with a Tukey post-hoc test. * $P < 0.05$; ** $P < 0.01$; *** $P < 0.001$.

- The mechanism by which FCS improves transdermal drug delivery has not been mentioned in the manuscript. It would be beneficial to include this information to provide a comprehensive understanding of the underlying mechanisms.

Reply: Thanks for point it out. The mechanism has been discussed in the manuscript as following:

“Conclusively, FCS, as the derivative of chitosan, could also enlarge cellular space by stimulating the phosphorylation of MLC, a phenomenon also observed for unmodified chitosan^{31,32}. The phosphorylation of MLC could then lead to the rearrangement of cytoskeletal structure, transforming tight junction related protein into the cytoplasm. Afterwards, the tight junction between epithelial cells would be opened and the intercellular space is enlarged to allow the permeation of our nanocomplexes. On the other side, the unique non-hydrophobic non-hydrophilic properties of fluorocarbon chains would make FCS less sticky when penetrating through various biological barriers^{41,42}. Therefore, FCS-based nanocomplexes also showed enhanced permeation via hair follicles through the transappendageal pathway.”

- Could the authors provide details on the loading capacity of the FCS-based system and the antibody ratio employed in the study? This information would contribute to a better understanding of the formulation and its effectiveness.

Reply: Thank you for the suggestion. The discussion of the loading capacity has been added in the manuscript as following:

“Dynamic light scatter (DLS) measurement (Figure S2) showed a single peak at ~200 nm for FCS/IgG nano-complexes, which were much larger than the sizes of free IgG, indicating that the majority of IgG should have been encapsulated by FCS.”

Figure S2. The DLS of FCS/IgG and free IgG

- Graphical representations of the western blot intensity would greatly enhance the visual presentation of the results. It is recommended that the authors include these graphical representations in the manuscript.

Reply: Thanks for point it out. The analysis of western blot intensities for MLC/pMLC have been added in the manuscript as Figure 2f.

Figure 2f. The relative intensity of MLC/pMLC with the addition of FCS/IgG.

- Have the authors investigated the toxicity of free chitosan and chitosan-based nanoformulations on normal healthy cells? It would be informative to include any relevant findings or considerations regarding the safety of these formulations.

Reply: Thank you for the advice. The cytotoxicity of chitosan (CS) and CS/IgG have been added in supporting figure S5 with discussion as following:

“As shown in Figure S4&S5, no obvious cytotoxicity of FCS and CS was observed for HACAT cells.

Figure S5. The relative viabilities of HACAT cells incubated with different concentrations of (a) CS or (b) CS/IgG nanocomplexes.

- In addition to surface marker analysis, functional assays on memory T cells could provide valuable insights into their effector functions in cancer immunity. These assays can assess the production of cytokines (e.g., interferon-gamma, interleukins) or cytotoxic activity against cancer cells. The authors may consider performing such functional assays to further support their conclusions.

Reply: Thank you for the suggestion. The cytokines analysis of memory T cells have been added in the supporting information with descriptions as following:

“On day 75 (pre boost) and day 90 (15 days after a single boost), cytokines in sera of mice after different administrations were measured (Figure S25). The levels of IL-12p40, an important marker of innate immunity, and IFN- γ , the typical markers of cellular immunity were all obviously higher in mice treated with FCS/S1/PolyIC, demonstrating that the transdermal delivery of vaccines would trigger long-term adaptive immune memory effect, which might be resulted from the long retention of co-delivered antigen and adjuvant.”

Figure S25. The sera cytokine level of IL-p40 and IFN- γ on (a&b) day 75 (pre boost) and (c&d) day 90.

8. Regarding the administration of nanoformulations to mice, it is important to determine the appropriate dosage and establish a standardized dosing regimen. Therefore, I have the following questions:

A. What is the specific dosage of nanoformulations being administered to the mice as part of the experimental procedure?

B. How was the dosing regimen for the nanoformulations standardized to ensure consistency and comparability across treatment groups?

Reply: Thank you for the advice. The dosage, application area and application time to the mice and rabbits were consistent during different experiments. The standardized application of the nanoformulation on mice and rabbits have been added as following:

“5. Preparation of FCS based transdermal ointment and in vivo administration

The synthesized FCS/IgG or other FCS/protein nanocomplexes were dropped onto the surface of blank ointment (Aquaphor®, mainly petrolatum) with a mass ratio at 1:1, such as 20 μ L FCS/protein with 20 mg ointment. Then, they were gently mixed up with each other. The transparent blank ointment would turn into milk-like ointment, which was then applied onto the mouse and rabbit skin for transdermal applications.

For mice administration, the mixed milk-like ointment was then swiped on the mouse skin with a round cover of 3.14 cm² (1 cm in radius) for 12 h. Transparent film (3M) was covered finally to avoid the influence of mice lick.

For rabbits administration, the mixed milk-like ointment was then swiped on the rabbit skin with a round cover of 50.24 cm² (4 cm in radius) for 12 h. Transparent film (3M) was covered finally to avoid the influence of rabbits lick.”

9. It would be valuable to include a discussion on the limitations associated with the use of aPD1/aPDL1 antibodies, particularly the risk of autoimmune diseases after intravenous injection. This information will provide context for the authors' decision to use transdermal delivery rather than intravenous administration. These details can be included in the introduction section.

Reply: Thanks for point it out. The discussion of the limitation of immune checkpoint antibodies was added in the introduction as following:

“With significantly enhanced therapeutic responses compared to systemic injection of antibodies at the same dose, our FCS-based local delivery of immune checkpoint antibodies to treat melanoma may lead to less concerns in systemic side effects considering the relatively low serum concentrations by the topical administration route.”

10. If authors can show the cell cycle analysis in the cells isolated from the tumor, that will ultimately add positives to the article.

Reply: Thank you for the suggestion. The main effect of the FCS-based system was to improve the

transdermal ability of biomacromolecules such as antibodies. The antitumor efficacy relied on the effect of antibody itself. Few studies have reported the influence of cell cycle by aPDL1 or aCTLA4.

11. Could the authors elaborate on how FCS promotes the phosphorylation of myosin light chain, inducing the contraction of actin and rearrangement of the cytoskeleton in HACAT cells?

Reply: Thank you for the advice. We are sorry for the limitation. The further detailed mechanism on how FCS promoted the phosphorylation was not explored in this work. The mechanism might be related to the property of chitosan, as chitosan by itself could promote the phosphorylation of myosin light chain. The related discussion was added in the manuscript as following:

“Conclusively, FCS, as the derivative of chitosan, could also enlarge cellular space by stimulating the phosphorylation of MLC, a phenomenon also observed for unmodified chitosan^{31,32}. The phosphorylation of MLC could then lead to the rearrangement of cytoskeletal structure, transforming tight junction related protein into the cytoplasm. Afterwards, the tight junction between epithelial cells would be opened and the intercellular space is enlarged to allow the permeation of our nanocomplexes.”

12. How was the involvement of the transappendageal pathway in FCS-based transdermal delivery determined? It would be beneficial to provide details on the experimental approach or techniques used to assess this pathway.

Reply: Thanks for point it out. The study of the transappendageal pathway was discussed in the manuscript as following:

“With the counter-staining of Keratin (Krt) 14, the hair follicles and sweat glands in deep dermis region were labeled. As shown in Figure 2g (white arrows), it was observed that FCS/IgG colocalized with hair follicles and sweat glands, indicating that the transappendageal pathway also played an important role in FCS-based transdermal delivery systems.”

Figure 2h. Representative immunofluorescence images exhibiting the colocalization of keratin 14 and FCS/IgG-Cy5.5 (white arrows).

13. Can the role of the transcellular route in the transdermal delivery of FCS-containing

nanocomplexes be further investigated using different experimental approaches or cell models? This could shed more light on the overall transdermal delivery mechanism.

Reply: Thank you for the advice. Another experiment about the transcellular route was conducted and added in the manuscript as following:

“Besides, the further evaluation was conducted by Franz diffusion system. As reported, exocytosis was usually mediated by clathrin. Therefore, skins were treated with 0.1 mg/mL chlorpromazine hydrochloride, the inhibitor for clathrin for 2 hours to compare the penetration ability with or without the clathrin inhibitor. As shown in Figure S8, the skin retention of FCS/IgG-FITC also showed no clear influence.”

Figure S8. The skin retention FCS/IgG-FITC for 12 hours with. Skin was treated with or without 0.1 mg/mL chlorpromazine hydrochloride (the inhibitor for clathrin) for 2 hours.

14. Were there any specific physicochemical properties of FCS that contributed to its ability to transport deeply into the hair follicles and permeate through the barriers within the follicles? Providing insights into these properties would enhance our understanding of the nanocomplex behavior.

Reply: Thank you for the suggestion. The discussion of the physicochemical properties of FCS was discussed in the manuscripts as following:

“Conclusively, FCS, as the derivative of chitosan, could also enlarge cellular space by stimulating the phosphorylation of MLC, a phenomenon also observed for unmodified chitosan^{31,32}. The phosphorylation of MLC could then lead to the rearrangement of cytoskeletal structure, transforming tight junction related protein into the cytoplasm. Afterwards, the tight junction between epithelial cells would be opened and the intercellular space is enlarged to allow the permeation of our nanocomplexes. On the other side, the unique non-hydrophobic non-hydrophilic properties of fluorocarbon chains would make FCS less sticky when penetrating through various biological barriers^{41,42}. Therefore, FCS-based nanocomplexes also showed enhanced permeation via hair follicles through the transappendageal pathway.”

Reviewer #3 (Remarks to the Author): with expertise in transdermal drug delivery (also for cancer immunotherapy)

In this study, fluorocarbon modified chitosan (FCS) for transdermal delivery of immune checkpoint blockade antibodies and SARS-CoV2 antigens was developed, and anti-cancer activity and antibody production were induced by the FCS delivery system. Thus, FCS would be interesting material for transdermal drug delivery. However, important information and data are lacking as follows.

1. Reviewer could not image the physicochemical and structural properties of FCS nanoparticles because no chemical structural information about FCS was provided in the manuscript. Therefore, it was difficult to judge whether FCS is a suitable material for transdermal drug delivery.

Reply: Thanks for point it out. The chemical structure of FCS was illustrated in Figure S1.

Figure S1. The ^{19}F NMR spectra and chemical structure of FCS.

2. Direct evidence of intradermal penetration of FCS nanoparticles should be shown. For example, confocal laser scanning microscopic images of skin section after treatment with FCS nanoparticles including fluorescence dye should be shown. Reviewer could not judge whether the proteins penetrate into skin as FCS nanoparticles.

Reply: Thank you for the suggestion. The confocal microscopy images of skin section with FITC-FCS/OVA-Cy5.5 have been added in the manuscript as following:

“Afterwards the skins in the Franz diffusion system post transdermal delivery was sliced for confocal microscopy with FITC-FCS/OVA-Cy5.5 to further evaluated the penetration of FCS nanocomplexes. As shown in Figure 1n, the colocalization of FCS/OVA was observed in the dermis region of the skin, demonstrating that FCS/OVA could penetrate into the skin dermis in the nanocomplex forms. In contrast, no obvious skin penetration was observed in the free OVA group.”

Figure 1n. Representative confocal images of mice skin treated with FITC-FCS/OVA-Cy5.5 for 12 h. Scale bar: 200 nm.

Figure 1n. Representative confocal images of mice skin treated with FITC-FCS/OVA-Cy5.5 for 12 h. Scale bar: 200 nm.

3. The phenomenon induced such as tight junction opening by FCS treatment are shown. However, the reason why FCS causes such phenomenon have not been examined.

Reply: Thank you for the advice. We are sorry for the limitation. The further mechanism on how FCS promoted the phosphorylation was not explored in this work. The mechanism might rely on the property of chitosan itself. The related discussion was added in the manuscript as following:

“Conclusively, FCS, as the derivative of chitosan, could also enlarge cellular space by stimulating the phosphorylation of MLC, a phenomenon also observed for unmodified chitosan^{31,32}. The phosphorylation of MLC could then lead to the rearrangement of cytoskeletal structure, transforming tight junction related protein into the cytoplasm. Afterwards, the tight junction between epithelial cells would be opened and the intercellular space is enlarged to allow the permeation of our nanocomplexes.

4. There are two “Figure 5”.

Reply: Thank you for the advice. The typo error has been corrected.

5. In later Figure 5 (maybe Figure 6)i, the photographic image should be upside down. In the figure, the stratum corneum (SC) is stained with DAPI, even though the SC has no nuclei. Also, the Cy5.5 and DAPI images of FCS/IgG completely overlap. Therefore, the reviewer could not believe these data.

Reply: Thank you for the suggestion. The image has been upside down for clear expression. In the figure, the skin sections with all layers were stained with DAPI, the outer layer of DAPI-stained cells should be keratinocytes in basal layers in epidermis instead of the stratum corneum. The misunderstandings have been changed.

Figure 6i Representative confocal fluorescence images to show the permeation depth FCS/IgG-Cy5.5 through the porcine skin in 12 h. Free IgG-Cy5.5 was used as a control in those experiments. Scale bar: 100 μ m.

Reviewer #4 (Remarks to the Author): with expertise in transdermal drug delivery, vaccines, immunology

In this manuscript, the authors present a fluorocarbon-modified chitosan (FCS) that enhances transdermal delivery of therapeutic proteins (antibodies and antigens) and nucleotides (PolyIC) to murine, rabbit, and porcine skin. This study builds on previous work from the group in which they developed the FCS polymer and evaluated its application for transmucosal delivery and treatment of bladder cancer (ref 25). In the current manuscript, the authors nicely demonstrate that this nanocomplex forming material can deliver checkpoint blockade antibodies (anti-PDL1 +/- anti-CTLA4) to treat cutaneous melanomas in mice. They also use the FCS to formulate transdermal vaccines delivering SARS-CoV-2 S1 or OVA antigens plus PolyIC adjuvant to immunize mice and rabbits. This approach and findings, especially with evidence of applicability in multiple animal models for multiple disease indications, would be relevant to a broad audience in the fields of drug delivery and biomaterials. Addressing the following comments will help to improve the manuscript for readers of Nature Communications.

1. Some key information is missing from the methods section. In particular, please add methods regarding the processing (enzymatic dissociation) of skin and tumors for flow cytometric analyses of T cells and DCs. Which Ki67 and IFN γ antibodies, as well as protein transport inhibitors and/or intracellular/intranuclear staining buffers, were used? How were S1-specific antibody titers calculated? The methods only mention measurement of 1:100 dilutions of serum, not titers. How was delivery of 125 I-IgG quantified in tumors and normalized?

Reply: We greatly appreciate the reviewer's suggestion. The detailed descriptions of experimental method have been added in the supporting information as following,

Process: "Spleens, tumors, skins and lymph nodes were collected and homogenized into single cell

suspensions following the standard protocol. Briefly, they were processed through mechanical disruption before digestion for 1 h at 37°C in an enzymatic solution with RPMI-1640 (10%FBS and 1%PS), 1.5 mg/ml collagenase IV(Sigma), 1.5 mg/ml collagenase I (Sigma), 1.5 mg/ml hyaluronidase (Sigma) and 0.2 mg/ml DNase I (Sigma). The samples were then passed through 200-mesh nylon mesh filters to obtain single-cell suspensions. The obtained single cell suspensions were incubated with anti-CD16/32 for 30 min at 4 °C. Then, these cells were stained with different antibodies according to the standard protocol.”

Ki67 and Foxp3:” To further analyze helper T cells in tumors, the suspension was stained with anti-CD3-FITC (Biolegend, Cat. 100203), anti-CD4-APC (Biolegend, Cat. 100411), anti-Foxp3-PE (Biolegend, Cat. 126403) antibodies with eBioscience™ Foxp3 / Transcription Factor Fixation/Permeabilization Concentrate and Diluent according to the manufacturer’s protocols. Briefly, after the cells were stained with CD16/CD32 and cell surface markers, they were fixed with the Foxp3 Fixation/Permeabilization working solution for 30 min at room temperature. Then, they were washed with permeabilization buffer at 500g for 5 minutes and stained with CD16/CD32 in the permeabilization buffer for 30 min at 4 °C. Then, they were stained with anti-Foxp3-PE for another 30 min at room temperature and resuspended in the flow cytometry staining buffer for analysis. For the analysis of cytotoxic T cell lymphocytes, the suspension was stained with anti-CD3-APC (Biolegend, Cat. 100206), anti-CD8a-PE (Biolegend, Cat. 100707), anti-CD45-PerCP (Biolegend, Cat. 103130) and anti-Ki67-FITC (Biolegend, Cat. 652410) for the level of Ki67 with the same protocol as the Foxp3. The suspension was also stained with anti-CD45-FITC (Biolegend, Cat. 157214), anti-CD8a-PE (Biolegend, Cat. 100707), anti-GranzymeB-PE-Cy7 (Biolegend, Cat. 372214) and anti-CD3-APC (Biolegend, Cat. 100206) for the level of Granzyme B with eBioscience™ Intracellular Fixation & Permeabilization Buffer Set.”

Antibody titer:” The S1 specific antibody titer was measured by a standard indirect ELISA (iELISA) method. Briefly, ninety-six-well ELISA plates (Nunc, Thermo) were coated overnight with 1 ng per well of the S1 protein in coating buffer. The plates were then blocked with 200 µl of ELISA assay buffer (Thermo) and incubated at room temperature for 2 hours. Each serum sample was tested in duplicate at a dilution from 1:100 to 1:640000 in twice in ELISA assay buffer (Thermo), in which 100 µl was then added into the wells of each plate for 2 h incubation at room temperature. The horseradish peroxidase (HRP)-conjugated goat anti-mouse IgG (1:10000, Abcam) was added for 1 h incubation at room temperature. Then, 100 µl TMB solution (Thermo) was added into each well. After 15 min incubation, 100 µl 2 M H2SO4 solution was added to stop the reaction. The absorbance of each well was read by the at 450 nm wavelength (PerkinElmer). The threshold was defied as double of the average absorbance in untreated group. The last dilution of a sample which is just larger than the threshold is defined as the antibody titer of this sample.”

Radio-isotope:”For in vivo biodistribution of 125I-IgG assay, B16F10 melanoma tumor bearing mice were topically administrated with free 125I-IgG and FCS/125I-IgG ointment for different time points. Then, the major organs including liver, spleen, kidney, heart, lung and tumor were collected and measured by Wizard2 Gamma Counter (PerkinElmer). The cumulation efficacy was calculated by the formulation

$$\%ID/g = R_n/R_0/m_n \times 100\%$$

Where R_n was the radio-intensity of the organ at the exact time point, m_n was the mass of the organ, R_0 was the radio-intensity of the applied ointment.”

2. Clear Granzyme B, Ki67, and IFN γ positive T-cell populations do not appear in the representative flow plots in Figures S12, S13, S14, and S18. How were the positions of these gates determined? Were FMO isotype controls used? Also, what type of ex vivo stimulation, protein transport inhibition, and/or intranuclear or intracellular staining buffers were used? Note that detection of Ki67, a nuclear antigen, requires fix/perm buffers optimized for intranuclear staining (as for FoxP3). Detection of IFN γ typically requires treatment with protein transport inhibitor and antigen-specific (peptide/protein) or non-specific (PMA/Ionomycin) stimulation.

Reply: Thanks for point it out. The representative flow plots and gating strategies have been added in the supporting figure S19&S20.

Figure S19. The analysis of CD4+ T cells in the primary tumor. (a) The gating strategy of CD4+ T cells and Tregs. (b) Representative flow cytometric plots of CD4+ T cells in CD45+ T cells. (c) Statistical analysis of CD4+ T cells in the tumor cells. (d) Representative flow cytometric plots of regulatory T cells (Foxp3+) in CD4+ T cells. (e) Statistical analysis of regulatory T cells (Foxp3+) in CD4+ T cells. Data are presented as mean \pm standard deviation ($n=4$).

Figure S20. The analysis of CD8+ T cells in the primary tumor. (a) The gating strategy of CD8+ T cells. (b) Representative flow cytometric plots of CD8+ T cells in CD45+ T cells. (c) Statistical analysis of CD8+ T cells in the tumor cells. (d) Representative flow cytometric plots of cytotoxic CD8+ T cells (Ki67+) in CD8+ T cells. (e) Statistical analysis of cytotoxic CD8+ T cells (Ki67+) in CD8+ T cells. Data are presented as mean \pm standard deviation ($n=4$).

FMO isotype controls and staining buffers have been added in the supporting information as following:

“Spleens, tumors, skins and lymph nodes were collected and homogenized into single cell suspensions following the standard protocol. Briefly, they were processed through mechanical disruption before digestion for 1 h at 37°C in an enzymatic solution with RPMI-1640 (10%FBS and 1%PS), 1.5 mg/ml collagenase IV(Sigma), 1.5 mg/ml collagenase I (Sigma), 1.5 mg/ml hyaluronidase (Sigma) and 0.2 mg/ml DNase I (Sigma). The samples were then passed through 200-mesh nylon mesh filters to obtain single-cell suspensions. The obtained single cell suspensions were incubated with anti-CD16/32 for 30 min at 4 °C. Then, these cells were stained with different antibodies according to the standard protocol.

To further analyze helper T cells in tumors, the suspension was stained with anti-CD3-FITC (Biolegend, Cat. 100203), anti-CD4-APC (Biolegend, Cat. 100411), anti-Foxp3-PE (Biolegend, Cat. 126403) antibodies with eBioscience™ Foxp3 / Transcription Factor Fixation/PermeabilizationConcentrate and Diluent according to the manufacturer’s protocols. Briefly, after the cells were stained with CD16/CD32 and cell surface markers, they were fixed with the Foxp3 Fixation/Permeabilization working solution for 30 min at room temperature. Then, they were washed with permeabilization buffer at 500g for 5 minutes and stained with CD16/CD32 in the permeabilization buffer for 30 min at 4 °C. Then, they were stained with anti-Foxp3-PE for

another 30 min at room temperature and resuspended in the flow cytometry staining buffer for analysis.

For the analysis of cytotoxic T cell lymphocytes, the suspension was stained with anti-CD3-APC (Biolegend, Cat. 100206), anti-CD8a-PE (Biolegend, Cat. 100707), anti-CD45-PerCP (Biolegend, Cat. 103130) and anti-Ki67-FITC (Biolegend, Cat. 652410) for the level of Ki67 with the same protocol as the Foxp3. The suspension was also stained with anti-CD45-FITC (Biolegend, Cat. 157214), anti-CD8a-PE (Biolegend, Cat. 100707), anti-GranzymeB-PE-Cy7 (Biolegend, Cat. 372214) and anti-CD3-APC (Biolegend, Cat. 100206) for the level of Granzyme B with eBioscience™ Intracellular Fixation & Permeabilization Buffer Set.”

The stimulation of IFN-γ+ T cells in spleen and tumors were added as following:”For the analysis of IFN-γ+ T cells in spleens, the suspension was firstly incubated with a stimulation medium with RPMI-1640 (10%FBS and 1%PS), 50 μM HEPES solution (Sigma), 1 mM Sodium pyruvate solution (Sigma), 50 μM 2-Mercaptoethanol, 10 μg/mL SI or OVA protein, and 1 × eBioscience™ Brefeldin A solution for 6 hours and then stained with anti-CD3-FITC (Biolegend, Cat. 100203), anti-CD4-PerCP (Biolegend, Cat. 100432), anti-CD8a-PE (Biolegend, Cat. 100707) and anti-IFN-γ-APC (Biolegend, Cat. 505810) with eBioscience™ Intracellular Fixation & Permeabilization Buffer Set. For the analysis of IFN-γ+ T cells in tumors, eBioscience™ Cell Stimulation Cocktail (plus protein transport inhibitors) (500X) was added in the stimulation medium to replace protein and Brefeldin A in the above protocol.”

3. In Figure 5i, it would be helpful to include representative fluorescence microscopy images from untreated control skin to confirm that the apparent Cy5.5 signal is not background autofluorescence. Also, while the caption says, “The white line represents the stratum corneum layer (SC),” no such line is apparent in the images.

Reply: Thank you for the suggestion. Another fluorescence microscopy experiment was conducted with the blank group. The confocal microscopy of skin section with FITC-FCS/OVA-Cy5.5 has been added in the manuscript as following:

“Afterwards the skins in the Franz diffusion system post transdermal delivery was sliced for confocal microscopy with FITC-FCS/OVA-Cy5.5 to further evaluated the penetration of FCS nanocomplexes. As shown in Figure 1n, the colocalization of FCS/OVA was observed in the dermis region of the skin while no obvious penetration was observed in both blank and free OVA group, which showed that FCS/OVA could penetrate into the skin dermis region as nanocomplex forms.”

Figure 1n. Representative confocal images of mice skin treated with FITC-FCS/OVA-Cy5.5 for 12

h. Scale bar: 200 nm.

4. For the Franz diffusion cell experiments used to quantify transdermal delivery in murine, rabbit, and porcine skin (Figures 1j-m, 5b-c, and 5g-h), it would be helpful to include in the figure captions the total dose per area that was applied to the skin, so delivery efficiency can be inferred. From the supplemental methods, it appears that a total dose of 200 $\mu\text{g}/\text{cm}^2$ was applied ($1 \text{ mg}/\text{mL} * 0.4 \text{ mL} / 2 \text{ cm}^2$). Also, since the FCS/protein formulations were applied to skin in PBS solution for the delivery experiments, compared to Aquaphor ointment with occlusive dressing for the in vivo experiments, the authors should briefly discuss how the vehicle and occlusive dressing could influence delivery efficiency and kinetics.

Reply: We greatly appreciate the reviewer's suggestion. The feeding doses in Franz diffusion cell experiments have been added in the figure captions.

Our FCS/protein formulations were applied onto the skin after mixing the solution with blank ointment, rather than in bare PBS solution. The addition of Aquaphor ointment was to maintain the moisture of the sample on the skin, otherwise the PBS solution containing FCS-based nanocomplexes would be quickly dried on the skin if without adding ointment. The following clarification has been added in the revised manuscript:

"For in vivo topical administration of our FCS-based nanocomplexes, blank ointment (Aquaphor®, mainly petrolatum) was mixed with the nanocomplex solution to maintain moisture for the sample applied on the skin (otherwise a bare buffer solution could be quickly dried after being applied on the skin)."

5. In the vaccine experiments, antigen-specificity of the T-cell populations was not evaluated by either peptide-MHC tetramer staining or ex vivo antigen stimulation and intracellular cytokine staining. As such, the reported changes in T-cell populations (IFN γ ⁺ and CD44⁺ CD62L⁻ effector memory T cells) cannot be attributed definitively to antigen-specific T-cell responses. The authors should acknowledge this limitation in the discussion.

Reply: Thank you for the suggestion. The IFN- γ ⁺ T cells was treated with stimulation medium for 6 h before staining. The detailed description of the experimental method has been added in the supporting information as following:

"For the analysis of IFN- γ ⁺ T cells in spleens, the suspension was firstly incubated with a stimulation medium with RPMI-1640 (10%FBS and 1%PS), 50 μM HEPES solution (Sigma), 1 mM Sodium pyruvate solution (Sigma), 50 μM 2-Mercaptoethanol, 10 $\mu\text{g}/\text{mL}$ S1 or OVA protein, and 1 \times eBioscience™ Brefeldin A solution for 6 hours and then stained with anti-CD3-FITC (Biolegend, Cat. 100203), anti-CD4-PerCP (Biolegend, Cat. 100432), anti-CD8a-PE (Biolegend, Cat. 100707) and anti- IFN- γ -APC (Biolegend, Cat. 505810) with eBioscience™ Intracellular Fixation & Permeabilization Buffer Set. For the analysis of IFN- γ ⁺ T cells in tumors, eBioscience™ Cell Stimulation Cocktail (plus protein transport inhibitors) (500X) was added in the stimulation medium to replace protein and Brefeldin A in the above protocol."

While for the memory T cells population, mice were treated with another boost on day 75 before the memory T cell analysis on day 90, the discussion was added in the manuscripts as following: "Moreover, the percentage of both effector memory CD4+ and CD8+ T cells (CD3+CD4+CD44+CD62L- and CD3+CD8+CD44+CD62L-) in the spleen of mice treated with FCS/S1/polyIC was also dramatically increased on day 90 with a single boost on day 75 (Figure S24), while those in mice with subcutaneous administration of S1/PolyIC appeared to be much lower than those in the transdermal group. On day 75 (pre boost) and day 90 (15 days after a single boost), cytokines in sera of mice after different administrations were measured (Figure S25). The levels of IL-12p40, an important marker of innate immunity, and IFN- γ , the typical markers of cellular immunity were all obviously higher in mice treated with FCS/S1/PolyIC, demonstrating that the transdermal delivery of vaccines would trigger long-term adaptive immune memory effect, which might be resulted from the long retention of co-delivered antigen and adjuvant."

Figure S24. The long-term adaptive immune effect of mice measured on day 90 after different vaccination administrations. Splenic lymphocytes were collected 90 days post administrations and stained with FITC-CD3, PE-CD44, APC-CD62L and PERCP-CD4 or PERCP-CD8. The CD44+CD62L- cells in both CD4+ and CD8+ T cells are effector memory CD4+ and CD8 T+ cells, respectively.

Figure S25. The sera cytokine level of IL-p40 and IFN- γ on (a&b) day 75 (pre boost) and (c&d) day 90.

- Page 2, “To realize transdermal delivery of biomacromolecule drugs especially proteins, novel chemical enhancers such as membrane penetrating peptides, as well as various physical enhancement devices including cavitation ultrasound, electroporation, thermal ablation, microdermabrasion and microneedles have been developed 11–16.” – References 11-13 describe transdermal delivery of small molecule drugs, not biomacromolecules. Consider replacing these references with others that focus on transdermal delivery of macromolecules, including: spherical nucleic acids (e.g., Zheng et al. PNAS 2012, DOI: 10.1073/pnas.1118425109) and STAR particles (e.g., Tadros et al. Nat Med 2020, DOI: 10.1038/s41591-020-0787-6).

Reply: Thanks for point it out. The related references have been updated as recommendation.

- Page 2, “Meanwhile, physical enhancement devices such as electroporation and sonophoresis not only can hardly be self-operated...” – There are multiple recent examples of handheld electroporation devices for cutaneous vaccination, including: Xia et al. PNAS 2021, DOI: 10.1073/pnas.211081711 and Smith et al. Nat Commun 2020, DOI: 10.1038/s41467-020-16505-0.

Reply: Thank you for the advice. The related references have been updated following your recommendation.

- Page 3, “...Aquaphor® as a cream formulation...” – Aquaphor is an ointment, not a cream, which is relevant to the current studies, as ointments and creams, which contain less oil, tend to have different rates of absorption into the skin. References to “cream” here and in the methods should be corrected.

Reply: Thank you for the suggestion. The description of cream has been changed into ointments.

9. Page 4, “As shown in Figure 1f, FCS/IgG showed similar CD spectrum to that of free IgG, indicating that the structure of protein remained nearly unchanged during after formation of such FCS-containing nanocomplexes. Similar result also was found in the comparison between FCS/OVA and free OVA in Figure 1g.” – The CD spectra in Fig 1f and 1g exhibit noticeable differences from ~200-215nm for protein vs protein + FCS formulations. Can the authors comment on the interpretation of these spectra differences?

Reply: We greatly appreciate the reviewer’s suggestion. The CD spectra showed different intensity with the same peak from 200-215 nm. The difference on intensity might result from the background of the solvent on short wavelength. Since they showed the same characteristic peak, they were regarded as unchanged structure. The discussion of the different from 200 nm to 215 nm has been added in the manuscript as following:

“As shown in Figure 1f, FCS/IgG showed similar CD spectrum characteristic peaks at around 202 nm, 206 nm and 216 nm to that of free IgG, indicating that the structure of protein remained nearly unchanged during formation of such FCS-containing nanocomplexes. The intensity difference on 200 nm might result from the solvent.”

10. Page 9, “...after the second i.v. injection of aPDL1 + aCTLA4, half of mice in this group died...” – Is this commonly observed in mice for the given dose of aPDL1 and aCTLA4? If so, please provide a reference.

Reply: Thank you for the advice. In our previous study, we have observed severe side effect of i.v. injection of aPDL1 & aCTLA4 in a dosage of 10 µg aPDL1 and 10 µg aCTLA4 for each mouse¹. Therefore, in this article, 20 µg aPDL1 and 20 µg aCTLA4 might cause acute cytokine storms so that 2 of 5 mice died after the second injection. On the other hand, combination therapy of aPDL1 & aCTLA4 was more studied in clinical experiments with reported severe side effects²⁻⁵.

11. Page 11, “Up to now, more than 180 SARS-CoV-2 vaccines are under research worldwide, and 26 SARS-CoV-2 vaccines are in the stage of clinical trials 48. However, all these vaccines need subcutaneous or intramuscular injection...” – These statistics are outdated (reference 48 is from 2020). There are currently multiple SARS-CoV-2 vaccines in clinical trials which are administered by intranasal route, and a number of microneedle-based COVID-19 vaccines in pre-clinical studies.

Reply: Thank you for the suggestion. The introduction has been up to date as following:

“Up to now, 242 SARS-CoV-2 vaccine candidates are in clinical development and 11 COVID-19 vaccines have been granted an Emergency Use Listing (EUL) by the WHO⁵¹.”

12. Pages 11-12, “Interestingly, mice with transdermal delivery of FCS/S1/polyIC showed almost similar antibody titer to that of mice with s.c. injection of S1/polyIC in two weeks, indicating that the transdermal delivery of FCS/S1/PolyIC could result in almost the same B cell activation level

compared with s.c. administrated vaccines (Figure 5g).” – Measurement of S1-specific total IgG binding antibody titers alone does not support the claim about “B cell activation levels.” Additional analyses of isotype switching, somatic hypermutation, and affinity maturation, or direct analysis of germinal center B cells would be needed to make this claim.

Reply: Thank you for the advice. The levels of B cells were not measured in the manuscript. Similar descriptions have been correlated as following:

“Interestingly, mice with transdermal delivery of FCS/S1/polyIC showed almost similar antibody titer to that of mice with s.c. injection of S1/polyIC in two weeks, indicating that the transdermal delivery of FCS/S1/PolyIC could result in almost the same level of humoral immunity compared with s.c. administrated vaccines (Figure 5g).”

13. Figures 5g and 6e – Since titers are presented on a log scale, bars and error bars should represent geometric mean and geometric standard deviation, respectively, instead of arithmetic mean and standard deviation or standard error.

Reply: Thanks for pointing it out. The statistical analysis has been correlated to geometric mean and geometric standard deviation.

Figure 5g. SARS-CoV-2 specific IgG antibody titers at different time intervals determined by ELISA.

Figure 6e. OVA specific IgG antibody titers in rabbit sera at different time intervals determined by ELISA.

14. Statistics are missing in the graphs in Figure 6; however, the corresponding caption has multiple references to statistics and statistical analyses.

Reply: We greatly appreciate the reviewer's suggestion. The statistical analysis has been added in Figure 6.

Figure 6. Evaluation of transdermal protein ability on rabbit and porcine models. (a) Schematic illustration of in vivo vaccination on the rabbit model. (b&c) Cumulative percentages of penetration (b) and skin retention (c) of FCS/IgG-FITC permeated across the rabbit skin over time. Total dosage: 200 µg/cm² (d) Schematic illustration of the experimental design showing transdermal delivery of OVA vaccination in rabbit model. (e) OVA specific IgG antibody titers in rabbit sera at different time intervals determined by ELISA. (f) Schematic illustration of ex vivo skin penetration on the porcine model. (g&h) Cumulative percentages of penetration (g) and skin retention (h) of FCS/IgG-FITC permeated across the porcine skin over time. Total dosage: 200 µg/cm² (i) Representative confocal fluorescence images and (j) statistical analysis to show the permeation depth FCS/IgG-Cy5.5 through the porcine skin in 12 h. Free IgG-Cy5.5 was used as a control in those experiments. Scale bar: 100 µm. All illustrations were created with BioRender.com. Data are presented as mean ± standard deviation (n=3 or 4). Statistical significance was calculated via one-way ANOVA with a Tukey post-hoc test. *P < 0.05; **P < 0.01; ***P < 0.001.

15. Survival analysis (Figure S10) should include a statistical analysis (logrank test).

Reply: Thank you for the advice. The statistical analysis has been added.

*Figure S14. Survival of mice in different groups. Mice were regarded as dead when they were truly dead or their tumor size was larger than 1,000 mm³. Statistical significance was calculated via log-rank (Mantel-Cox) test. * $P < 0.05$; ** $P < 0.01$; *** $P < 0.001$.*

16. In Figure S16, the y-axis for the second row of flow plots should be “FoxP3” instead of “Treg.”

Reply: Thank you for the suggestion. The typo error has been corrected.

Figure S19. Representative flow cytometric plots of (Figure 4j) regulatory T cells in the distant tumors after different treatments.

17. Page 4, “...both FCS/IgG and at 1:1 showed sizes around 100 nm...” – The sizes determined by TEM and DLS appear to be ~ 200 nm.

Reply: Thank you for the advice. The discussion has been correlated.

18. Page 5, “...when the amount of IgG increased, the zeta potential of FCS/IgG nanocomplex turned

to be negative with aggregation...” – According to Fig 1c, the lowest measured zeta potential was still positive (+6.97 mV), so not “negative”, but “less positive.”

Reply: Thanks for pointing it out. The description has been correlated.

19. Page 5, “In this case, human skin dermis cells HACAT were used...” – HaCaT cells are epidermal cells.

Reply: Thank you for the suggestion. The description has been correlated.

20. Figure 1j-m – The figure caption refers to “cumulative percentage”; however, the graphs show “Skin Permeation” and “Retention” in terms of $\mu\text{g}/\text{cm}^2$. Should the caption say “cumulative permeation” instead?

Reply: Thank you for the advice. The description has been correlated.

21. Figure 3 caption – “(a) Schematic illustrations illustrating the localized transdermal administration of FCS/aPDL1 & FCS/aCTLA4 for the treatment of B16F10 melanoma tumors.” – FCS/aCTLA4 is not shown in Figure 3a or anywhere in Figure 3.

Reply: We greatly appreciate the reviewer’s suggestion. The typo error has been corrected.

Reference

1. Zhu, W. *et al.* Oral Delivery of Therapeutic Antibodies with a Transmucosal Polymeric Carrier. *ACS Nano* **17**, 4373–4386 (2023).
2. Davies, M. & Duffield, E. A. Safety of checkpoint inhibitors for cancer treatment: strategies for patient monitoring and management of immune-mediated adverse events. *Immunotargets Ther* **Volume 6**, 51–71 (2017).
3. Boutros, C. *et al.* Safety profiles of anti-CTLA-4 and anti-PD-1 antibodies alone and in combination. *Nat Rev Clin Oncol* **13**, 473–486 (2016).
4. Childers, B. G. *et al.* Operative management of immune checkpoint colitis following in-transit melanoma: Case report. *Front Oncol* **13**, (2023).
5. De Velasco, G. *et al.* Comprehensive Meta-analysis of Key Immune-Related Adverse Events from CTLA-4 and PD-1/PD-L1 Inhibitors in Cancer Patients. *Cancer Immunol Res* **5**, 312–318 (2017).

REVIEWERS' COMMENTS

Reviewer #1 (Remarks to the Author):

The authors have responded well to reviewers' comments. I think the revised manuscript meets the requirements for publication in Nature Communications.

Reviewer #2 (Remarks to the Author):

In the present study, the authors developed a novel method for delivering a promising cancer immunotherapy drug, aPDL1, through the skin to directly treat melanomas. The researchers used a nanocomplex formulation containing aPDL1 and demonstrated that it was effective in reducing tumor growth in mice. The formulation was able to penetrate the skin through paracellular and trans appendageal routes, and the treatment resulted in enhanced antitumor immune responses with no apparent toxicity.

I have reviewed the manuscript and find the research to be valuable and worthy of publication. The authors have performed satisfactory revisions to the manuscript. The manuscript in the current form may be accepted.

Reviewer #3 (Remarks to the Author):

The reviewer was convinced by the author's responses.

Reviewer #4 (Remarks to the Author):

The authors have addressed each of my comments from the first review. I have a few minor comments on the revised sections:

- 1) Figure S19b - Should the y-axis (APC) label be CD4 and the x-axis (FITC) label be CD3?
- 2) Figures S19c,e and S20c,e captions say "Statistical analysis of..." but the four corresponding figures present bar graphs without any statistics.

3) Figure 1n - "Scale bar: 200nm" - should this be "200 μ m" instead of "200nm"?

REVIEWER COMMENTS

Reviewer #1 (Remarks to the Author):

The authors have responded well to reviewers' comments. I think the revised manuscript meets the requirements for publication in Nature Communications.

Reviewer #2 (Remarks to the Author):

In the present study, the authors developed a novel method for delivering a promising cancer immunotherapy drug, aPDL1, through the skin to directly treat melanomas. The researchers used a nanocomplex formulation containing aPDL1 and demonstrated that it was effective in reducing tumor growth in mice. The formulation was able to penetrate the skin through paracellular and trans appendageal routes, and the treatment resulted in enhanced antitumor immune responses with no apparent toxicity.

I have reviewed the manuscript and find the research to be valuable and worthy of publication. The authors have performed satisfactory revisions to the manuscript. The manuscript in the current form may be accepted.

Reviewer #3 (Remarks to the Author):

The reviewer was convinced by the author's responses.

Reviewer #4 (Remarks to the Author):

The authors have addressed each of my comments from the first review. I have a few minor comments on the revised sections:

1) Figure S19b - Should the y-axis (APC) label be CD4 and the x-axis (FITC) label be CD3?

Reply: Thanks for point it out. The labelling error has been corrected.

2) Figures S19c,e and S20c,e captions say "Statistical analysis of..." but the four corresponding figures present bar graphs without any statistics.

Reply: Thanks for the suggestion. The captions have been changed as following:

"(e) Data analysis of regulatory T cells (Foxp3+) in CD4+ T cells."

"(e) Data analysis of cytotoxic CD8+ T cells (Ki67+) in CD8+ T cells."

3) Figure 1n - "Scale bar: 200nm" - should this be "200µm" instead of "200nm"?

Reply: Thanks for point it out. The typo error has been corrected.